# Modelling debris transport within glaciers by advection in a full-Stokes ice flow model

Anna Wirbel[1], Alexander Helmut Jarosch[2], and Lindsey Nicholson[1]

[1]Institute of Atmospheric and Cryospheric Sciences, University of Innsbruck, Innsbruck, Austria
[2]Institute of Earth Sciences, University of Iceland, Reykjavík, Iceland

**Abstract.** Glaciers with extensive surface debris cover respond differently to climate forcing than those without supraglacial debris. In order to include debris-covered glaciers in projections of glaciogenic runoff and sea-level rise, and to understand the paleoclimate proxy recorded by such glaciers it is necessary to understand the manner and timescales over which a supraglacial debris cover develops. Because debris is delivered to the glacier by processes that are heterogeneous in space and time, and these debris inclusions are altered during englacial transport through the glacier system, correctly determining where, when, and how much, debris is delivered to the glacier surface requires knowledge of englacial transport pathways and deformation. To achieve this, we present a model of englacial debris transport in which we couple an advection scheme to a full-Stokes ice flow model. The model performs well in numerical benchmark tests, and we present both 2D and 3D glacier test cases that, for a set of prescribed debris inputs, reproduce the englacial features, deformation thereof, and patterns of surface emergence predicted by theory and observations of structural glaciology. In a future step, coupling this model to a (i) debris-aware surface mass-balance scheme and (ii) supraglacial debris transport scheme will enable the co-evolution of debris-cover and glacier geometry to be modelled.

## 1 Introduction

All mountain glaciers carry rock and dust material within the ice. This can originate from gravitational mass movements from the surrounding valley walls, aeolian deposition, or basal erosion (Benn and Evans, 2010). Rock and dust debris deposited onto the surface of a glacier in the accumulation zone is buried by subsequent snowfall and transported englacially with the glacier ice as it flows downslope. In the ablation zone of a glacier, ice flow transports debris towards the glacier surface and surface ice ablation leaves behind a residue of rock material (Fig. 1a). If debris supply and ablation is sufficiently high, and transport of rock material out of the glacier system is inefficient, a debris-covered glacier can develop, where a large portion of the ablation zone is covered with a continuous layer of rock material (Kirkbride, 2011).

A surface debris cover more than a few centimeters thick inhibits surface ablation of ice and thus alters glacier runoff, local water resources and contribution to sea level change. It also affects glacier dynamics and geometry such that low-angled, stagnating debris-covered ice can survive for longer at lower altitudes than neighbouring clean ice glaciers (Benn et al., 2012; Anderson and Anderson, 2016). Thus the paleoclimatic signal represented by sediment deposits from a debris covered-glacier is not the same as one from a clean ice glacier.

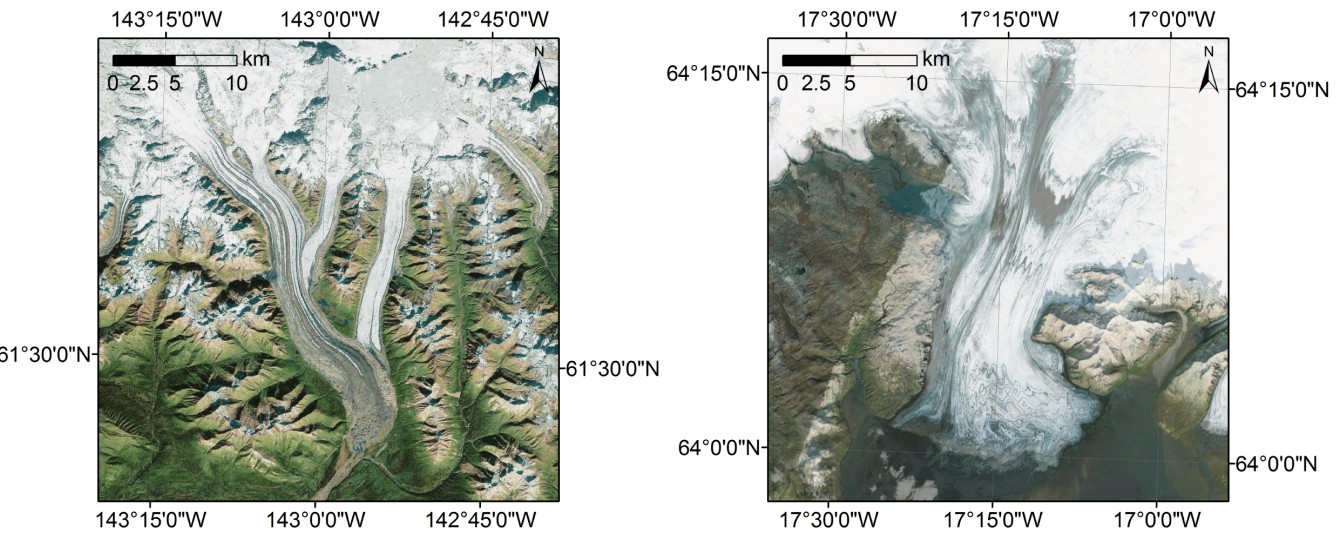

**Figure 1.** (a) Debris-covered Kennicott Glacier, Wrangell Mountains, Alaska, USA and (b) deformed englacial bands emerging at the surface of Skeiðarárjökull, an outlet glacier of Vatnajökull ice cap, Iceland. Source: ESRI basemap DigitalGlobe imagery.

Fluxes of ice and debris change over time in response to climatic variations, and in space due to differences in local site characteristics. Kirkbride (1989) proposes that variations in ice mass influx serve to unify a process-continuum of deformational geomorphological features of mixed ice and debris composition. The implication of this process-continuum is that glaciers can transition between rockglaciers, debris-covered glaciers and clean ice glaciers through space or time as a result of the varying

5 ice influx (Kirkbride, 1989; Ackert, 1998; Clark et al., 1998). Accordingly, the extent of a debris cover varies inversely with the glacier mass balance, whereby the debris cover extent is governed by transport dominant conditions (higher ice velocities and lower surface ablation) during periods of positive mass balance and ablation dominant conditions (lower ice velocities and higher ablation) during periods of negative mass balance (Kirkbride, 2000). In addition, debris covers can be formed instantaneously by isolated events such as ash fall or a large rockfall onto the glacier ablation zone (e.g. Nield et al., 2013;

10 Hewitt, 2009; Shugar et al., 2012; Reznichenko et al., 2011). The complex interplay between debris supply and ice supply and their variation in space and time mean that the thickness and character of the debris cover, and its resultant impact on the glacier behaviour is also strongly space and time-dependent. In order to resolve this, it is necessary to understand how the debris co-evolves with the glacier. This requires tackling many component parts to model the full debris-covered glacier system. To date, existing numerical models of debris-covered glaciers either restrict debris inputs to the ablation zone (Konrad

15 and Humphrey, 2000; Menounos et al., 2013; Vacco et al., 2010), prescribe an englacial debris concentration (Bozhinskiy et al., 1986) or use empirical relationships to describe accumulation of debris on the glacier surface (Jouvet et al., 2011).

Recent studies apply simplified treatment of englacial transport (Rowan et al., 2015; Anderson and Anderson, 2016), but as yet, no model explicitly resolves full 3D (three-dimensional), time evolving transport of debris within the ice flow field of the glacier body. This is a significant omission because, as surface debris mainly originates from localized debris inputs (rockfall or mixed avalanche events) in the accumulation zone, modelling englacial transport is crucial to predict the location and timing

of surface emergence of debris, as well as its concentration and its spatial extent, all of which are required to constrain the nature of the developing debris cover and its resultant impact on glacier behaviour.

Prevailing stress conditions, and the resulting strain and velocity fields, control sediment transport within an ice body. Individual clasts are considered to be predominantly passively transported by glacier ice, unless within the basal traction zone, and so their shape remains fundamentally unaltered by transport. For a static velocity field, the pathway of such a feature is

identical to a streamline within the glacier, but with the evolving glacier geometry these will change. However, debris inputs from rock, mixed snow/ice avalanches and other gravitational mass movements tend to be deposited as bodies of polymictic ice-sediment mixtures, which become severely deformed in the course of transport through the glacier, as revealed by studies of structural glaciology (e.g. Fig. 1b, Jennings et al., 2014; Mackay et al., 2014). Hence, the initial shape of the deposit will be changed significantly, and this englacial deformation will affect the pattern of debris emergence at the glacier surface (e.g.

Goodsell et al., 2005). In order to numerically model transport and deformation of sediment inclusions, a full representation of 3D velocity fields resolving all spatial gradients is essential, which calls for a full-Stokes ice flow modelling approach.

Here we present a new model that simulates transport, and resultant deformation, of material within a glacier coupled to 3D resolved ice flow, and we demonstrate the capabilities and performance of the model through a series of evaluation simulations. Although dynamics of debris-ice mixtures can differ from clean ice dynamics depending on several parameters

such as concentration of debris, particle size and temperature (Moore, 2014), in this work we assume that sediment inclusions within the glacier do not affect ice rheology due to the small total amount of transported material in comparison to overall ice volume. The model is coded in python and relies on the FEniCS framework, an open-source software for automated solution of partial differential equations (Alnæs et al., 2015; Logg et al., 2012a). The model employs an existing benchmarked full-Stokes ice flow model (*icetools*, Jarosch, 2008, now implemented in FEniCS) to compute 3D velocity fields that govern an advection

algorithm used to describe debris transport.

Assuming that ice is an incompressible fluid, and consequently that the ice flow fields must be divergence-free, any deformational patterns inducing horizontal elongation, must, at the same time, cause vertical compression. In the context of englacial debris transport, this implies that the initial debris concentration is constant for an initial control volume of ice being tracked (i.e. seen from Lagrangian perspective). The incompressibility assumption demands also the absolute values of concentration to

remain constant during transport when following the initial control volume of ice as it is becoming deformed during transport. To solve the transport problem mathematically, we take an Eulerian approach. The accuracy of the results is directly related to mesh size. If the mesh size was chosen infinitesimally small, the concentration features would recover, over the entire transport path, their initial values and sharp layers at their boundaries (if initial debris inputs are delineated by sharp boundaries). As a consequence of a fixed mesh, the bigger the mesh size, the greater the amount of numerical smearing in the simulations which

results in a decrease of maximum and marginal concentration values and a smearing towards the edges of the concentration

features. This problem is inherent to the method chosen but can be dealt with by applying an appropriate mesh size, such that numerical smearing is minimized according to the application of interest.

The model presented here forms part of an envisaged fully-integrated model framework that, by incorporating a (1) free surface evolution scheme including debris-aware mass balance subroutines and (2) transport model for debris at the glacier
surface interacting with the mass balance subroutines, will be capable of simulating the transient response of debris-covered glaciers, with predetermined debris inputs, to a changing climate.

This paper is structured as follows: Sect. 2 provides details about the equations governing ice flow and how the transport problem is addressed from a mathematical perspective. Sect. 3 describes the numerical schemes employed. Sect. 4 details the test simulations performed, the results of which are presented in Sect. 5 and discussed in terms of model performance,
limitations and applicability in Sect. 6. Conclusions and outlook are presented in Sect. 7.

## 2   Mathematical formulation

### 2.1   Full-Stokes formulation for ice flow

Ice is treated as an incompressible, non-linear viscous fluid, whose velocity and pressure distribution can be described by the incompressible stationary Stokes equations on a spatial domain $\Omega \epsilon \mathbb{R}^3$, representing the ice body:

$$-\nabla \cdot [\eta(\nabla \mathbf{u} + (\nabla \mathbf{u})^\top)] + \nabla p = \rho_{\text{ice}}\mathbf{g} \qquad \text{in } \Omega \qquad (1a)$$

$$\nabla \cdot \mathbf{u} = 0 \qquad \text{in } \Omega. \qquad (1b)$$

Here $\mathbf{u}$ is the 3D velocity field, $\eta$ is the non-linear viscosity, $p$ is the pressure, $\rho_{\text{ice}}$ is the density of ice and $\mathbf{g}$ is the acceleration due to gravity. The density of ice is assumed to be constant in time and space. By including the standard rheology of ice (Glen, 1955; Nye, 1957), its non-linear viscosity can be described by:

$$\eta = \frac{1}{2}A^{\frac{-1}{n}}\dot{\epsilon}^{\frac{(1-n)}{n}}, \qquad (2)$$

where $\varepsilon = 1/2(\nabla \mathbf{u} + (\nabla \mathbf{u})^\top)$ is the strain rate tensor and $\dot{\epsilon} = \sqrt{0.5\varepsilon_{ij}\varepsilon_{ji}}$ is the effective strain rate. $A$ represents the Glen rate factor and $n$ the Glen flow law exponent. The computational domain $\Omega$ is confined by a free surface boundary at the ice-air interface ($\partial\Omega_{\text{top}}$) satisfying:

$$2\eta\varepsilon \cdot \mathbf{n} - p\mathbf{n} = 0 \quad \text{on } \partial\Omega_{\text{top}}, \qquad (3)$$

where $\mathbf{n}$ is the outward pointing surface normal. At the ice-bedrock interface ($\partial\Omega_{\text{bed}}$) either a no-slip (Dirichlet-) boundary condition (Eq. 4a) or, for glaciers where basal sliding contributes to total movement, an alternative ice-bedrock (Neumann-) boundary condition can be applied (Eq. 4b) in conjunction with an appropriate sliding law conditioning the interface parallel ice velocity components at the glacier base.

$$\mathbf{u} = 0 \quad \text{on } \partial\Omega_{\text{bed}} \qquad (4a)$$

$$\mathbf{u} \cdot \mathbf{n} = 0 \quad \text{on } \partial\Omega_{\text{bed}} \qquad (4b)$$

## 2.2 Advection of material within a glacier

To describe transport and associated deformation of advected material within a glacier, we employ the linear transient advection-diffusion equation:

$$\frac{\partial c}{\partial t} = \nabla \cdot (D\nabla c) - \nabla \cdot (\mathbf{u}c) + r \quad \text{in } \Omega \tag{5a}$$

$$c = 0 \quad \text{on } \partial\Omega_0, \tag{5b}$$

where $c$ is the concentration of the material, $D \geq 0$ is the diffusion coefficient, $\mathbf{u}$ is the divergence-free velocity field and $r$ represents any internal sources or sinks. In the case of transport of debris through a glacier, it is reasonable to assume $r = 0$ and that material is predominantly transported by advection, therefore we currently neglect diffusion by setting $D$ sufficiently small. As we focus on englacial transport in this study, we set $c = 0$ at the domain boundaries except the parts of the boundary where an input location is assigned ($\partial\Omega_0$). Based on these assumptions and defining a constant diffusion coefficient $D$, Eq. 5 becomes:

$$\frac{\partial c}{\partial t} = D\nabla^2 c - \mathbf{u} \cdot \nabla c \quad \text{in } \Omega \tag{6a}$$

$$c = 0 \quad \text{on } \partial\Omega_0. \tag{6b}$$

At starting time $t_0$, a known initial concentration is given for all grid points on locations ($\mathbf{x}$) by:

$$c(\mathbf{x}, t = t_0) = c_0(\mathbf{x}) \quad \text{in } \Omega. \tag{7}$$

Apart from an initial concentration, material can enter the domain at the boundaries as a single input at time $t_{\text{input}}$ or by a defined rate as a function of time and location.

## 3 Numerical schemes and model software

The model consists of three main components, (1) ice deformation (*icetools*), (2) adaptive mesh refinement according to concentration patterns (*refine_gl*) and (3) debris transport (*advect_gl*). All model components are individual open-source modules coded in python and utilizing the FEniCS framework (Alnæs et al., 2015; Logg et al., 2012a). Computations are performed on unstructured meshes (triangles in 2D (two-dimensions) and tetrahedrons in 3D), that allow for variable mesh size according to local requirements in spatial resolution and geometry complexity. Computational meshes are generated with *gmsh* (Geuzaine and Remacle, 2009), an open-source finite element mesh generator.

## 3.1 FEniCS software

FEniCS is an open-source project designed for automated solution of partial differential equations (PDEs) by finite element methods (FEM) (https://fenicsproject.org, Alnæs et al., 2015; Logg et al., 2012a). It includes several components such as DOLFIN (Logg and Wells, 2010; Logg et al., 2012c), FFC (Kirby and Logg, 2006; Logg et al., 2012b; Ølgaard and Wells,

2010) and FIAT (Kirby, 2004, 2012), which enable automatic solution of linear and non-linear problems once the variational forms of the PDEs are expressed in the Unified Form Language (UFL, Alnæs et al., 2014; Alnæs, 2012).

## 3.2 Ice deformation

3D ice velocities are computed using *icetools*, a parallelized, open-source full-Stokes model for ice flow (Jarosch, 2008) that solves Eq. 1-4. A mixed function space of continuous piecewise quadratics and linears is used to compute ice velocity and pressure. The capability of the model to simulate 3D velocity fields for complex ice bodies has been demonstrated in previous studies (Jarosch, 2008; Jarosch and Gudmundsson, 2012). Initial versions of *icetools* accounted for stress-dependent ice viscosity using a Picard-iteration scheme, but here we employ an updated version, where the non-linear problem of including stress-dependent viscosity is solved by the Newton-Method.

## 3.3 Mesh refinement

Deline et al. (2015) collated statistics of the dimensions of deposits from massive rock slope failures onto glaciers documented since 1900. For the documented events, they found mean deposit lengths (n $=55$) and final thickness (n $= 20$) of 6.3 (min. 18.2/ max. 1.4) km and 3.5 (min. 22.0/ max. 1.0) m respectively. These values represent the upper limits on the likely dimensions of individual rockfall events onto glaciers, although megaslides could have larger dimensions. In order to resolve debris or ash deposits covering the range of these observations, computational meshes are required to have a spatial resolution in the sub–meter scale. In the case of simulating glaciers several kilometers long, this would lead to immense computational costs. Therefore, we take the approach of refining the mesh locally, i.e. only those areas where concentration is present. In order to avoid mesh refinement at every computation time step, we increase the area of refinement by a spatial radius $R_{\mathrm{cells}} = \mathbf{u}_{\mathrm{max}} dt_{\mathrm{ADV}} c_{\mathrm{ref}}$ surrounding the concentration features, where $\mathbf{u}_{\mathrm{max}}$ is the maximum velocity in the refined area, $dt_{\mathrm{ADV}}$ is the refinement time step (see Sect. 3.5) and $c_{\mathrm{ref}}$ is a positive defined constant. In this manner, the mesh is refined in an area that covers the actual concentration feature and the distance it can possibly be transported within the refinement time step.

For 2D simulations, mesh refinement is implemented entirely in python using the FEniCS software framework (Alnæs et al., 2015; Logg et al., 2012a). A function representing the coordinates (Fig. 2b) where the concentration exceeds a threshold (Fig. 2a) and $R_{\mathrm{cells}}$ is defined on the domain-wide coarse mesh (Fig. 2c). Using this function, the cells to be refined can be marked at any stage of mesh refinement. The marked cells are refined uniformly until all affected cells have an area smaller than a threshold $c_{\mathrm{vol}}$ (Fig. 2d). In this study, $c_{\mathrm{vol}}$ is set to $0.075 \ \mathrm{m}^2$ (equivalent to an equilateral triangle of edge length $0.416$ m), which, according to the findings in Sect. 5.1, is suitable to successfully represent englacial debris features originating from surface layer deposits of several meters in thickness. For 3D simulations, *gmsh* (Geuzaine and Remacle, 2009) is used to create a new refined mesh at every refinement time step. Therefore, a domain-wide coarse mesh is updated with information on the coordinates where concentration exceeds a threshold. To create the new mesh, the cell size within the radius $R_{\mathrm{cells}}$ of these coordinates is set to $L_{\mathrm{csize}}$. This parameter is representative of the average cell size within this area. In order to further reduce the number of required cells, the mesh is primarily refined in streamline direction. Therefore, the coordinate points are first shifted using the present velocity and a time step of $0.75 \ dt_{\mathrm{ADV}}$. As the refinement is based on the same domain-wide coarse

mesh for each refinement time step, both approaches (2D and 3D mesh refinement) prevent over-refinement and at the same time, the need for mesh coarsening.

Adaptive mesh refinement strategies often employ *a posteriori* error estimation (e.g. John, 2000). The PDE is solved and the assigned error estimators and indicators are used to mark the cells for refinement and potentially coarsening. Subsequently, the marked cells become modified and the PDE is solved on the newly refined mesh. This process is repeated until the error estimators and indicators fall below a user-defined tolerance within every cell. This procedure can require many iterations, resulting in high computational costs. In the case of debris inclusions within glaciers, we deal with rather smooth concentration fields, except for sparse areas of high concentration that often, initially or over time, exhibit a band-like shape. Therefore, rather than focusing only on sharp layers that are the main contributors to high errors on too-coarse meshes, we perform the refinement on the entire area of high concentration. Instead of using error estimators and indicators to locate the cells for mesh refinement, our methods use a concentration-threshold, cell-based refinement indicator. This indicator is computed just once per refinement time step and all affected cells and those within a velocity-based distance ($R_{\mathrm{cells}}$) are (a) refined until a problem-specific mesh size tolerance ($c_{\mathrm{vol}}$) is achieved (2D) or (b) are created with a cells size prescribed by $L_{\mathrm{csize}}$ (3D) . The results of the benchmark test in Sect. 5.1 derived with our approach compare well with that derived using adaptive mesh refinement based on *a posteriori* error estimation (de Frutos et al., 2014), this demonstrates that our approach is an acceptable balance between accuracy and savings in computational costs. The 3D benchmark test in Sect. 5.1, demonstrates the suitability of the 3D mesh refinement approach.

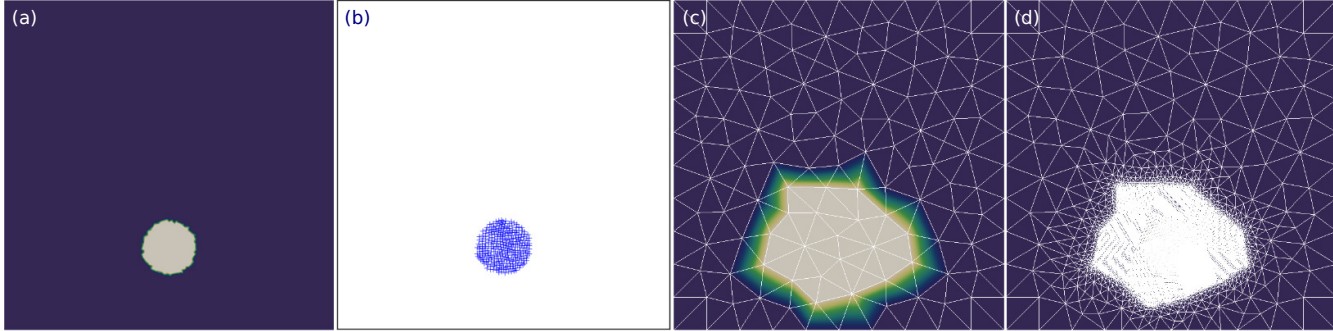

**Figure 2.** Illustration of mesh refinement. (a) Initial concentration field where bright colours indicate high concentration values. (b) Coordinate points of grid locations where concentration exceeds a threshold of 0.01. (c) Function that shows high values, indicated by bright colours, at all cells of the coarse mesh that lie within the radius $R_{\mathrm{cells}}$ of the respective coordinate points in (b). (d) Representation of the final refined mesh.

### 3.4 Material transport

The transient advection-diffusion equation is discretized in time by an implicit Euler scheme and a standard continuous Galerkin FEM is used for the space discretization, separating the temporal and spatial discretizations. The concentration is expressed as a scalar function in a continuous piecewise linear function space. In the case of advection-dominated transport,

solving Eq. 6 via standard continuous Galerkin FEM leads to non-physical spurious oscillations (e.g. Bochev et al., 2004). In order to inhibit these spurious oscillations and ensure stability, we employ the Streamline-Upwind-Petrov-Galerkin (SUPG) approach (Hughes and Brooks, 1982). In this method, a residual-based stabilization term is added to the variational form and in this way introduces artificial diffusion to the system in streamline direction. The stabilization term is based on the the residual of Eq. 6 including the time derivative and a mesh-size dependent stabilization parameter $\tau$. Following John and Novo (2011) and Bochev et al. (2004), in the advection-dominated case (i.e. Peclet numbers greater than 3) we use a stabilisation term of $\mathcal{O}(h_K)$ and define it as $\tau = \frac{h_K}{2||\mathbf{u}||}$, where $h_K$ is a measure of the local cell size and $\mathbf{u}$ is the divergence-free velocity field.

## 3.5 Time stepping

The refinement time step prescribes how often the refinement has to be performed and, in conjunction with the velocity field, defines the total number of cells in the mesh. For example, for a given englacial debris concentration, the total number of cells in the refined mesh increases with increasing refinement time step as the distance the debris inclusions are transported within this time step also increases. In order to minimize computational effort, the refinement time step has to be chosen according to the characteristics of the computer system that is used to run the computations.

The computation time step for the advection module is derived using the Courant-Friedrich-Lewy condition, applied on the smallest cell size and the maximum velocity within the refined region. In this study, we apply Courant numbers ranging from $0.5 - 1.5$. The work of Bochev et al. (2004) combined with the tests in the supplementary material, show that the SUPG stabilisation scheme coupled to a Crank-Nicholson or an implicit Euler scheme for time-dependent advection-dominated, advection-diffusion problems is stable for this choice of Courant numbers.

## 4  Model simulations

Direct evaluation of our advection model against real world glacier cases is not possible at present because (i) comprehensive field measurements of englacial debris transport are not available and (ii) simulating the full glacier system would require further model development as outlined in Sect. 1.

Nevertheless, an evaluation of how well the model (a) performs and (b) reproduces structures observed in glaciers is important. To this end, we present results from specific numerical tests that benchmark the advection module, followed by glacier simulations for a 2D profile of an alpine valley glacier and an idealized 3D glacier geometry. In these glacier simulations the flow fields are computed by solving Eq. 1-4 for given geometries and are kept fixed as no mass balance routine is coupled to the model yet. The benchmark tests were performed to quantitatively ascertain that the numerics of our model adequately meet the requirements of the task in terms of mass conservation, numerical stability and prevention of non-physical spurious oscillations and numerical smearing. By comparing the results to those of the published tests, the suitability, stability and general performance of the advection module are evaluated. The glacier simulations are used to qualitatively evaluate how well the coupled iceflow - advection model reproduces glacial structures related to idealized debris inputs of various dimensions

illustratively representing rockfall deposits, extensive ash layer or debris avalanche deposits and crevasse-fill in comparison to the structures predicted by theory or observed in the field.

## 4.1 Benchmark tests

The numerical Examples 1 and 2 in Bochev et al. (2004) are chosen to demonstrate the effect of the SUPG stabilisation approach in terms of reducing non-physical spurious oscillations that are a known problem for standard continuous Galerkin FEM schemes in the case of advection-dominated problems, and the stability of this stabilisation scheme for a wide range of Courant numbers that control the time stepping. Details of this set of numerical tests are presented in the supplementary material and here we present only the results of the most demanding numerical test that we subjected the model to, which is Example 4 in de Frutos et al. (2014). This test is known as the "rotating three body problem" (LeVeque, 1996; John and Novo, 2011; de Frutos et al., 2014) and is a standard test for computing advection of a scalar quantity in an incompressible flow field using the transient advection-diffusion equation in the advection-dominated case. Furthermore, in the study of de Frutos et al. (2014), the capabilities of *a posteriori* error-based adaptive mesh refinement are evaluated. By comparing the published results to those reproduced with our implementation of adaptive mesh refinement, we evaluate our method. In the "rotating three body problem", a slotted cylinder, a hump and a conical body undergo clockwise rotation in a divergence-free velocity field. A visualisation of the velocity field is given in Fig. A4a in the supplementary material. In order to reproduce the results with our model, we set up the velocity field, initial and boundary conditions as described in Example 4 in de Frutos et al. (2014) for an advection-dominated case ($D = 10^{-6}$ m$^2$ s$^{-1}$). However, we employ the mesh refinement and time stepping described in Sect. 3 and redefine the model domain as $\Omega = (0, 100) \times (0, 100)$ meters. By doing this enlargement, the size of concentration features becomes comparable to the size of debris inputs in the glacier simulations. Initializing the mesh refinement with the same cell area threshold as is used in the glacier simulations, allows us to estimate the level of accuracy that we can achieve in the glacier cases. Here, we present results of computations using two different refinement time steps, (a) small refinement time step of $0.01\,\pi$ s and (b) a larger refinement time step of $0.1\,\pi$ s. This results in (a) 200 and respectively for (b), in 20 refinement time steps for a full rotation of $2\,\pi$ s ($t_{\text{total}}$). The computation time step is derived using a Courant number of $0.5$. In order to evaluate the chosen cell area threshold, we perform convergence tests where (a) $\|c_{\text{h}} - c_{\text{e}}\|_{\text{L}_2} = \sqrt{\sum_{k=1}^{n}(c_{\text{h}_k} - c_{\text{e}_k})^2}$ the L2 norm of the error between the computed finite element solution $c_{\text{h}}$ and the exact solution $c_{\text{e}}$, where $n$ is the number of computation locations, and (b) the Root Mean Square (RMS) error between the computed finite element solution and the exact solution for different cell size thresholds are computed. Therefore, we first compute the exact solution on the same mesh that is used in the finite element solution. To subject the model to an even more severe test, a second set of simulations is performed where the velocity field is prescribed as a swirling flow (LeVeque, 1996), but all other settings remain identical. Due to the swirling flow, the shapes of the three bodies become deformed but at total time ($t_{\text{total}}$, at $t = 1.5$ s), the three bodies recover their initial shape. An animation of the swirling flow is included in the supplementary material.

To test the model capabilities in 3D, we reproduce the numerical test described in Christensen (1993), where a sphere of high concentration undergoes rotation. In this test, the velocity field is constructed in a manner, that the shape of the sphere is deformed throughout the rotation, but after a full rotation of $2\pi$ s ($t_{\text{total}}$), the sphere recovers its initial shape. A

visualisation of the velocity field is given in Fig. A4b in the supplementary material. The model domain is defined as $\Omega = (0, 32) \times (0, 32) \times (0, 40)$ meters and the mesh refinement is initialized with $L_{\text{csize}} = 0.15$ m. The refinement time step is set to $0.04\,\pi$ s and the Courant number to $0.5$. By comparing the results to the analytical solution of the problem presented in Christensen (1993), model performance and chosen refinement settings can be evaluated.

## 4.2 Glacier tests

The purpose of these tests is to demonstrate the characteristics of debris transport within mountain glaciers, not to reproduce a particular event on a specific glacier. Hence, all velocity computations are initialized with a no-slip condition at the glacier-bedrock boundary, the flow law exponent n is set to 3 and the Glen rate factor A is set to $2.4 \times 10^{-24}$ s$^{-1}$Pa$^{-3}$, a standard value for temperate ice (Cuffey and Paterson, 2010). The density of ice $\rho_{\text{ice}}$ is set to $917\,\text{kg m}^{-3}$. The transport simulations are initialized with a debris concentration field $c_0$. X, Y, Z coordinates are used to identify a debris-deposition zone characterizing (a) a part of the glacier-atmosphere, glacier-bedrock or glacier-sidewall interface that receives instant, continuous or variable debris input or (b) a localized debris inclusion as e.g. a remnant of a rockfall event or a crevasse-fill. This is implemented by assigning desired values of initial concentration at the respective locations to the function $c_0$, which is set to $0$ everywhere else on the entire domain. In the presented glacier simulations, all debris inclusions have been deposited in a single event, hence they are all initialized as inclusions within the glacier, i.e. the entire glacier-atmosphere boundary belongs to $\Omega_0$. In the 3D cases, the concentration is initialized with a smoothed function at the boundaries of the feature. This is done to represent it most efficiently in a continuous function space. Debris concentration that is transported beyond the boundaries of the glacier domain is removed from the system. The concentration itself is a scalar function able to have arbitrary numbers. It can be converted into actual debris mass as a function of the actual debris density and concentration of the initial debris deposit, i.e. the percentage of debris *versus* ice or snow in the initial volume of the deposit. In this study, we present model simulations for initial debris concentrations of the value 100, that can be scaled according to the case-relevant initial proportions of debris and ice. For example, in the case of an ash layer deposit, the initial ash concentration will likely make up to almost 100 %, compared to a mixed avalanche deposit that is likely to have much lower initial concentrations of debris *versus* snow or ice.

### 4.2.1 2D glacier test

For the 2D glacier test simulation, a 100 m spatial resolution longitudinal profile of bedrock and glacier surface for Haute Glacier d'Arolla was downloaded from the Ice Sheet Model Intercomparison Project (ISMIP) website (http://homepages.ulb.ac.be/~fpattyn/ismip/). These data represent the glacier in 1930, based on digitization of data from (Blatter et al., 1998), further described in (Pattyn, 2002). The longitudinal profile of 1930 is 5 km long. Haute Glacier d'Arolla is an alpine valley glacier with supraglacial debris covering approximately 10% of the glacier surface in 2012 (Reid et al., 2012).

In this test, we apply idealized debris inputs in the glacier accumulation area and track the evolution of the internal debris concentration while being transported in a fixed velocity field. For the 2D glacier profile, horizontal and vertical velocity components are shown in Fig. 3. In the supplementary material, Fig. A6 shows the surface velocity computed with a Glen rate factor $A = 10^{-16}$ a$^{-1}$Pa$^{-3}$ as used in ISMIP, demonstrating that our model reproduces the ISMIP results in Pattyn et al.

(2008). Five debris features of different size and input location are prescribed at varying time intervals (Fig. 4). These debris deposits of varying size, shape and location of deposition were chosen to facilitate analysis of the interplay between debris input location, deformation during transport and the zone of emergence. At $t = 0$, the prescribed initial debris concentration field consists of a circular debris inclusion centered at $x = 500$ m and $z = 3052$ m with a radius of 25 m (C0), a group of three crevasse-fills of $2 - 5$ m width and $50 - 75$ m length between $x = 1800$ m and $x = 1900$ m (CRV) , and a ca. 4 m thick debris layer covering the glacier surface between $x = 296$ m and $x = 854$ m of its length (D1). Subsequently, further debris layers are prescribed as follows: after 25 years, a ca. 9.5 m thick layer is deposited on the glacier surface between $x = 1000$ m and $x = 1600$ m (D2), and after 50 years another layer of ca. 5 m thickness is deposited between $x = 300$ m and $x = 1950$ m (D3). The circular inclusion and the debris layer deposits were prescribed to provide tight constraints on the shapes, whereas the vertical inclusions were initialized having irregular shaped boundaries considered more representative for actual crevasse-fills of variably sized debris material. The entire simulation, displayed in the video of the supplementary material, covers 90 years. The mesh refinement is initialized with $c_{\mathrm{vol}} = 0.075$ m$^2$, the refinement time step is set to 0.2 years and the Courant number to define the computation time step is set to 0.5.

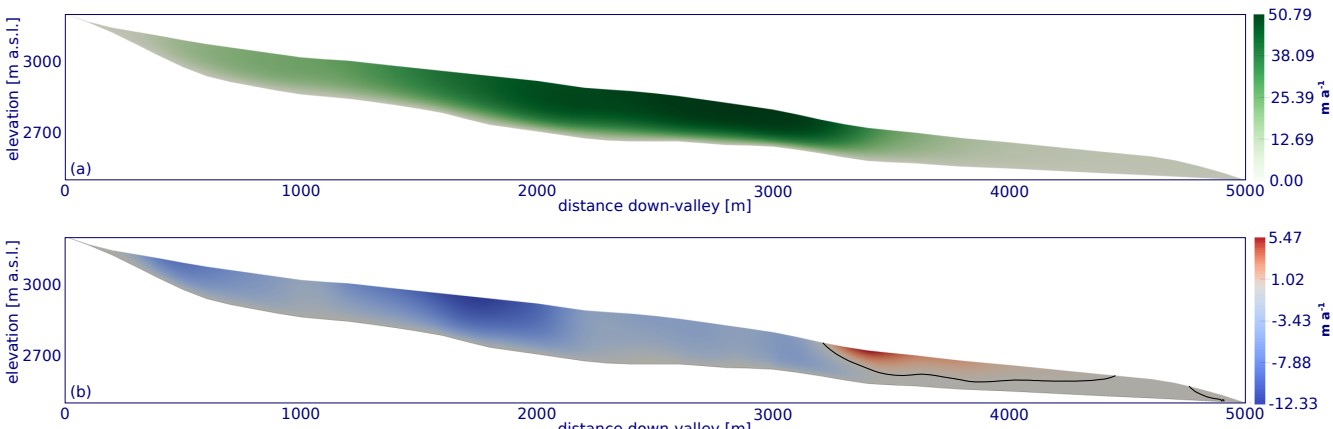

**Figure 3.** Velocity components computed with *icetools* for the 2D longprofile of Haute Glacier d'Arolla. (a) Horizontal velocity $(\mathrm{ma}^{-1})$ and (b) vertical velocity $(\mathrm{ma}^{-1})$ and contour line of zero vertical velocity in grey.

### 4.2.2 Idealized 3D glacier test

In the 3D glacier test, we perform simulations for an idealized glacier geometry. The geometry represents a valley glacier, including topographically-induced complexities such as a wide accumulation basin leading to a narrow valley, a bump in the bedrock geometry, as well as a turn of the valley itself. In this manner, topographic features that control ice flow in an alpine setting are represented, though idealized to reduce computational effort. The idealized ice geometry and the computed 3D

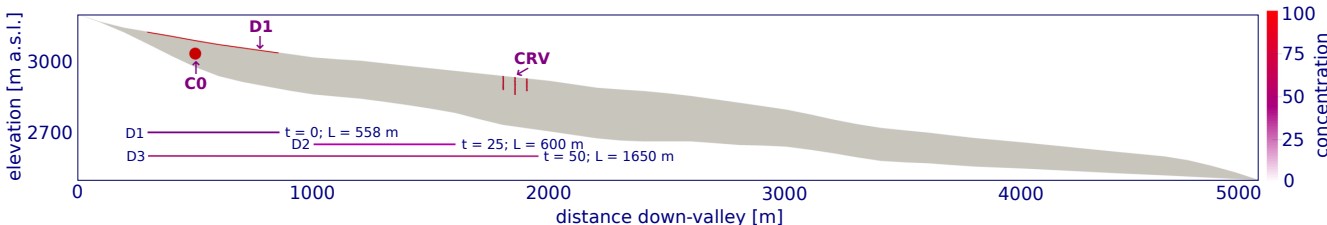

**Figure 4.** Debris concentration at time step $t = 0$ years, where C0 indicates the circular debris inclusion, D1-D3 the surface debris layer deposits and CRV the crevasse-fills. The horizontal lines indicate the location and extent of the additional debris layers deposited at the surface at the stated times. $L$ indicates the horizontal distance of the glacier surface where debris is deposited.

velocity field is shown in Fig. 5. In this test, the mesh refinement is initialized with $L_{\text{csize}} = 0.4$ m, the refinement time step is set to $0.4$ years and the Courant number to $1.5$.

The simulation is initialized with a spherical debris inclusion centered at $x = 0$ m, $y = 0$ m and $z = 270$ m in the accumulation area with a radius of $9.5$ m. This initial concentration is chosen to aid visualization of the transport and deformation rather than to best represent likely en- and supraglacial debris deposits.

## 5   Results

### 5.1   Benchmark tests

Our results of reproducing Examples 1 and 2 in the numerical results in Bochev et al. (2004) are shown in Figs. A2 and A3 in the supplementary material. These simulations demonstrate the efficiency of our SUPG algorithm implementation for reducing non-physical spurious oscillations in the solutions and allow us to choose suitable Courant numbers to ensure numerical stability.

The results of the "rotating three body problem" (de Frutos et al., 2014) are shown in Fig. 6, for the refinement time step $0.1\,\pi$ s. The results as well as animations for all sets of tests can be found in the supplementary material. In Fig. 6a-c, the initial condition, the solution after a full rotation of $2\pi$ s and the solution on the underlying mesh are shown. The shapes of the concentration features are well recovered in the case of both refinement time steps (see Fig. 6b for refinement time step $0.1\,\pi$ s and Fig. A5b in the supplementary material for refinement time step $0.01\,\pi$ s). Positive and negative oscillations in the solution are shown in Fig. 6d-e. The highest oscillations occur where the gradients in concentration are strongest, i.e. at the walls of the slotted cylinder. To measure the magnitude of remaining spurious oscillations in the solution, the difference of the maximum and minimum value of the solution is given in de Frutos et al. (2014). In our results, for case (a) $max(u) - min(u) = 1.2526$ with $87773$ cells in the final mesh, whereas for case (b) $max(u) - min(u) = 1.2524$ with $142792$ cells in the final mesh. These oscillations are slightly higher, but comparable to the values of $1.1010 - 1.1301$ reported in de Frutos et al. (2014). The total number of cells in the final meshes is larger in our computations, as the mesh refinement is performed in an interval of (a)

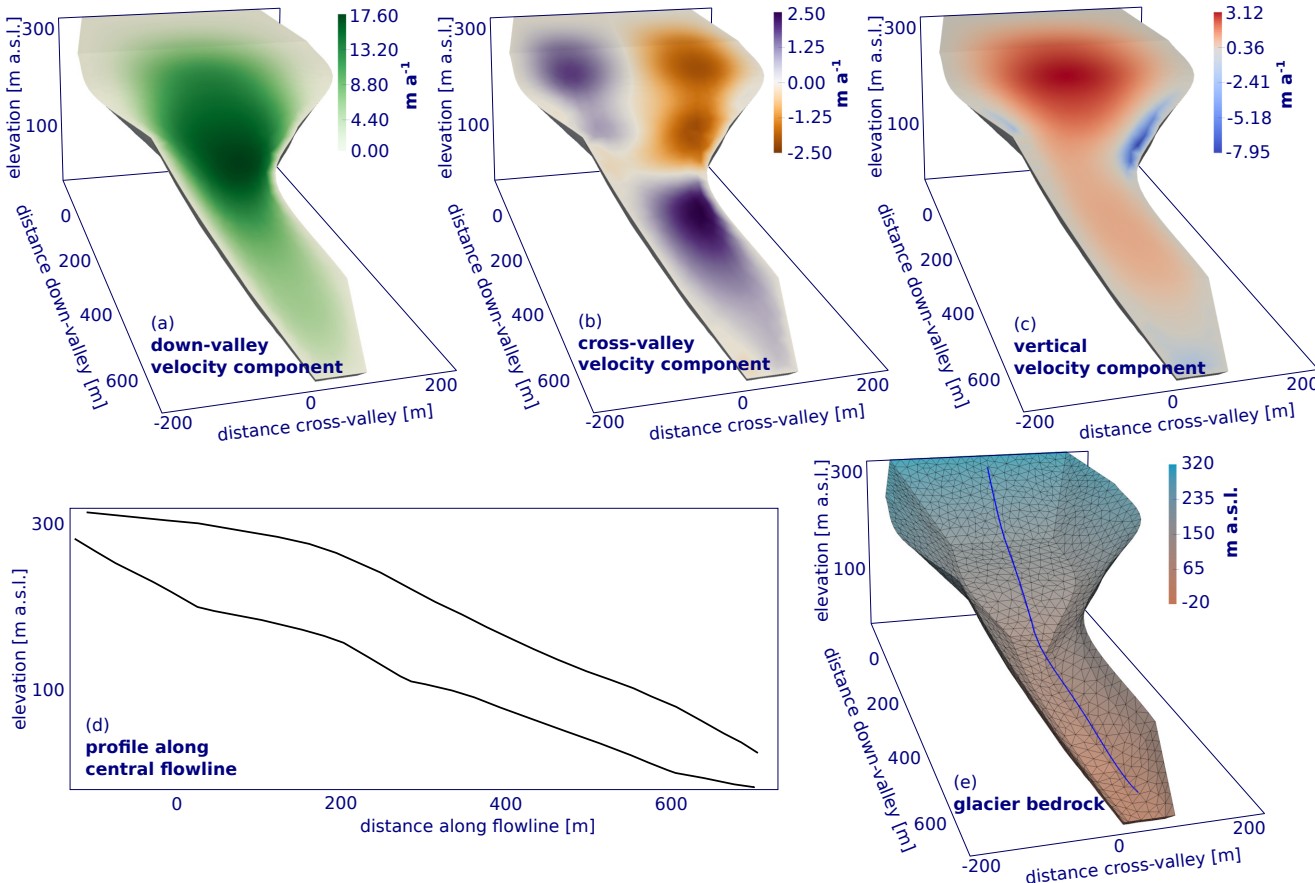

**Figure 5.** Idealized geometry and computed velocity components ($\mathrm{ma^{-1}}$) for the 3D glacier case. (a) Down-valley direction (along x-axis), (b) cross-valley direction (along y-axis) and (c) vertical direction (along z-axis), (d) 2D profile along a central flowline and (e) glacier bedrock elevation and flowline from (d) in blue.

$0.01\,\pi$ s or (b) $0.1\,\pi$ s and not individually for every computation time step. Mass loss is $< 0.009\%$ for both refinement time steps. Results of the convergence test for decreasing cell area thresholds, that are required to drive the mesh refinement, are shown in Fig. 6f. The chosen cell area threshold of $0.075\ \mathrm{m^2}$ yields acceptable results. A further decrease leads to a drastic increase in computational costs, with only a small increase in model accuracy. Also, when the initial concentration pattern is

5 subjected to a more complex, swirling flow (LeVeque, 1996), the results of these more challenging test simulations again show satisfactory model performance, as can be seen in Fig. 7 and Fig. A6 in the supplementary material.

The results of the 3D test following Christensen (1993) are illustrated in Fig. 8. An animation of the full rotation is given in the supplementary material. During the full rotation mass loss/gain is less than 0.01%. The shape of the concentration feature is in good agreement with the analytical solution as indicated by the isosurfaces of concentration (Fig. 8), just the

10 highest concentrations (isosurface 90) are not captured well due to numerical smearing. This also causes the maximum value

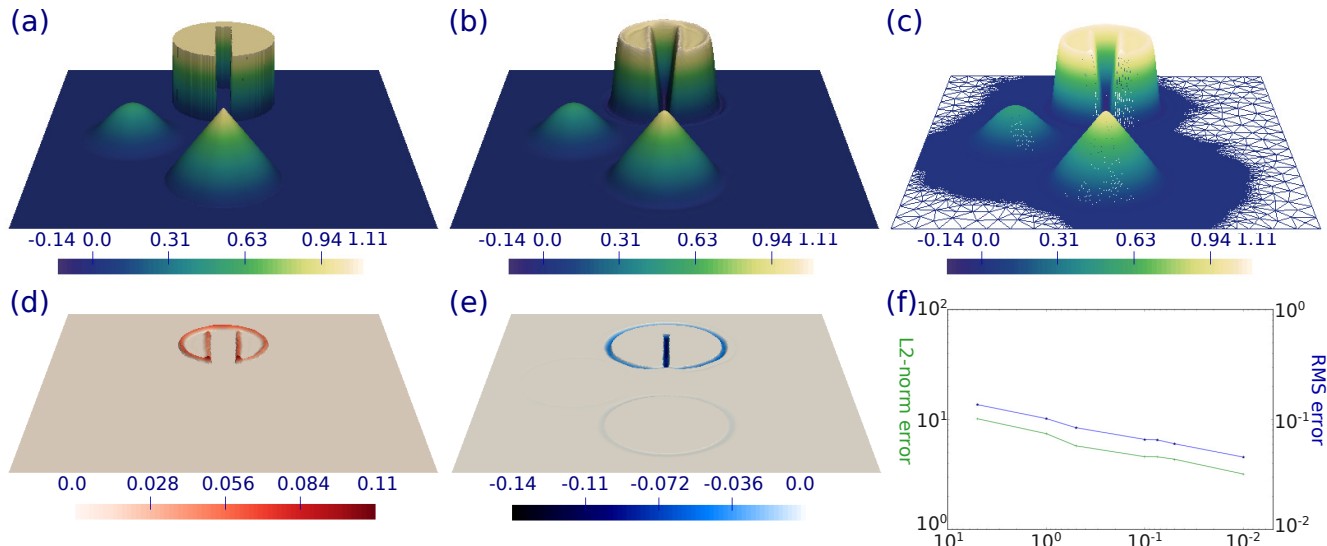

**Figure 6.** Results of 2D three body rotation test for refinement time step $0.1\pi$ s. (a) Initial condition, (b) solution after one full rotation and (c) solution after one full rotation on the underlying mesh. (d) Overshoots (values greater than maximum initial value, as deviation from $max(c_{initial}) = 1.0$) and (e) undershoots (values smaller than minimum initial value, as deviation from $min(c_{initial}) = 0.0$) at $t_{total}$. (f) Results of the convergence test as a function of mesh refinement parameter $c_{vol}$. Colour scales show concentration values.

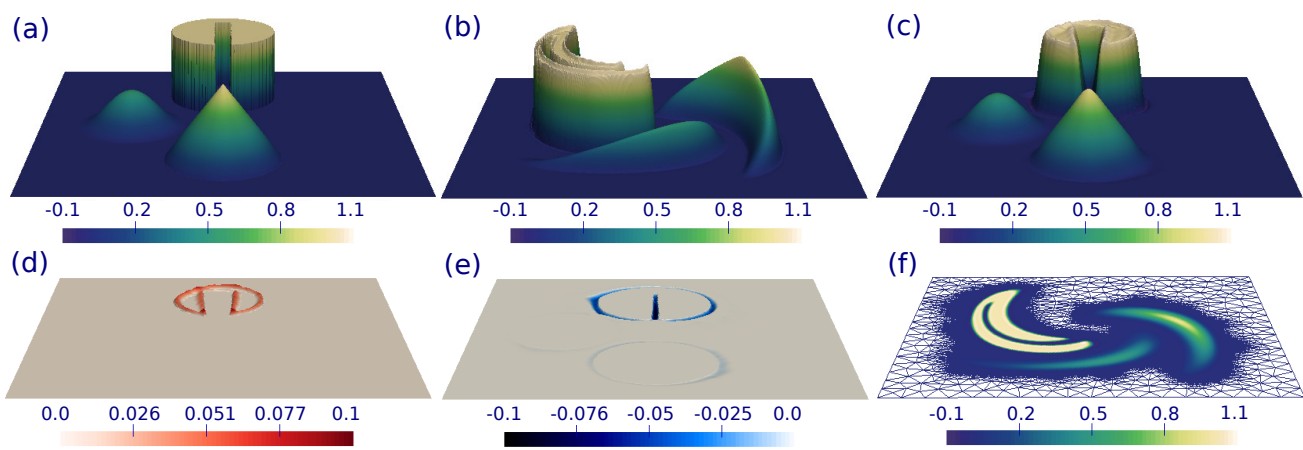

**Figure 7.** Results of 2D swirling flow three body test for refinement time step $0.01\pi$ s. (a) Initial condition, (b) solution at $t_{total}/2$ and (c) solution at $t_{total}$. (d) Overshoots (values greater than maximum initial value, as deviation from $max(c_{initial}) = 1.0$), (e) undershoots (values smaller than minimum initial value, as deviation from $min(c_{initial}) = 0.0$) at $t_{total}$ and (f) solution at $t_{total}/2$ on the underlying mesh, shown in plan view. Colour scales show concentration values.

of concentration in the final solution to reduce to 83 (initially 100) and the spreading of very low concentrations (< 5) over a larger volume compared to the analytical solution.

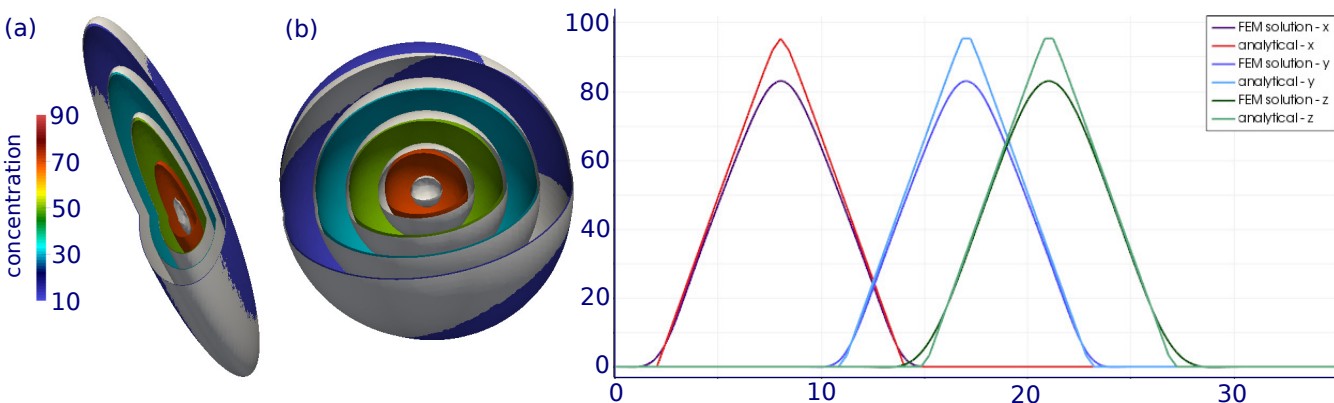

**Figure 8.** Results of 3D rotation test. Isosurfaces of concentration 10, 30, 50, 70, and 90 for the FEM solution (in colour) and the analytical solution (in solid grey) at (a) $t_{total}/2$ and at (b) $t_{total}$. Note that for the FEM solution the isosurface 90 is missing as the maximum values of concentration decreased to 87.5 (at $t_{total}/2$) and 83 (at $t_{total}$). (c) Cross profiles parallel to the x-, y- and z-axis for the analytical and the FEM solution at $t_{total}$.

## 5.2 2D glacier test

The upper boundary of the circular inclusion is initially located approximately 30 m beneath the glacier surface. During transport, it becomes severely elongated, as the upper part of the feature is transported faster with the ice flow than its lower part (Fig. 9a). After some travel time and sustained elongation, the vertical distance between the initially circular inclusion and the debris layer deposit D1 (Fig. 4) gradually decreases due to the vertical gradients in velocity (Fig. 9b). The initial surface emergence of the circular inclusion occurs later and further downglacier than any of the other imposed concentration features. It travels the longest distance and reaches the greatest depths within the glacier flow field.

The crevasse-fills are initially quasi-perpendicular to the glacier surface. As they are transported through the glacier, the vertical inclusions become deformed and exhibit a progressively more arcuate shape (Fig. 9d). These features reach the glacier surface and emerge first at $x = 2715$ m. They are removed from the glacier domain over the course of 31 years and over a distance of 580 m of the glacier surface. As they progressively emerge to the surface, the angle of outcrop rotates from vertical to upglacier dipping bands (Fig. 9d).

The layer-shaped debris inputs (D1-3 in Fig. 4) do not only have different characteristics such as length and thickness but are also deposited at different locations on the glacier surface. D1 first reaches the glacier surface at $x = 3450$ m and over a 90 year simulation period emerges over 109 m of the glacier surface. D2 first reaches the glacier surface at $x = 3190$ m and over a 90 year simulation period emerges over 281 m of the glacier surface. D3 first reaches the glacier surface at $x = 2645$ m and over a 90 year simulation period emerges over 645 m of the glacier surface. In comparison to D1 and D3, the upper limit

of D2 is located ca. 700 m further downglacier. The zone of emergence is significantly shorter, it becomes less elongated and is exhumed in a shorter period of time, compared to D1 and D3. Another characteristic, observed in the modelling results, is a reduction in the distance between the debris bands further downglacier, coinciding with decreasing ice velocities in this part of the glacier. The dip angle at the point of emergence to the surface differs between the three debris layer deposits and also changes as each layer feature is advected further downglacier.

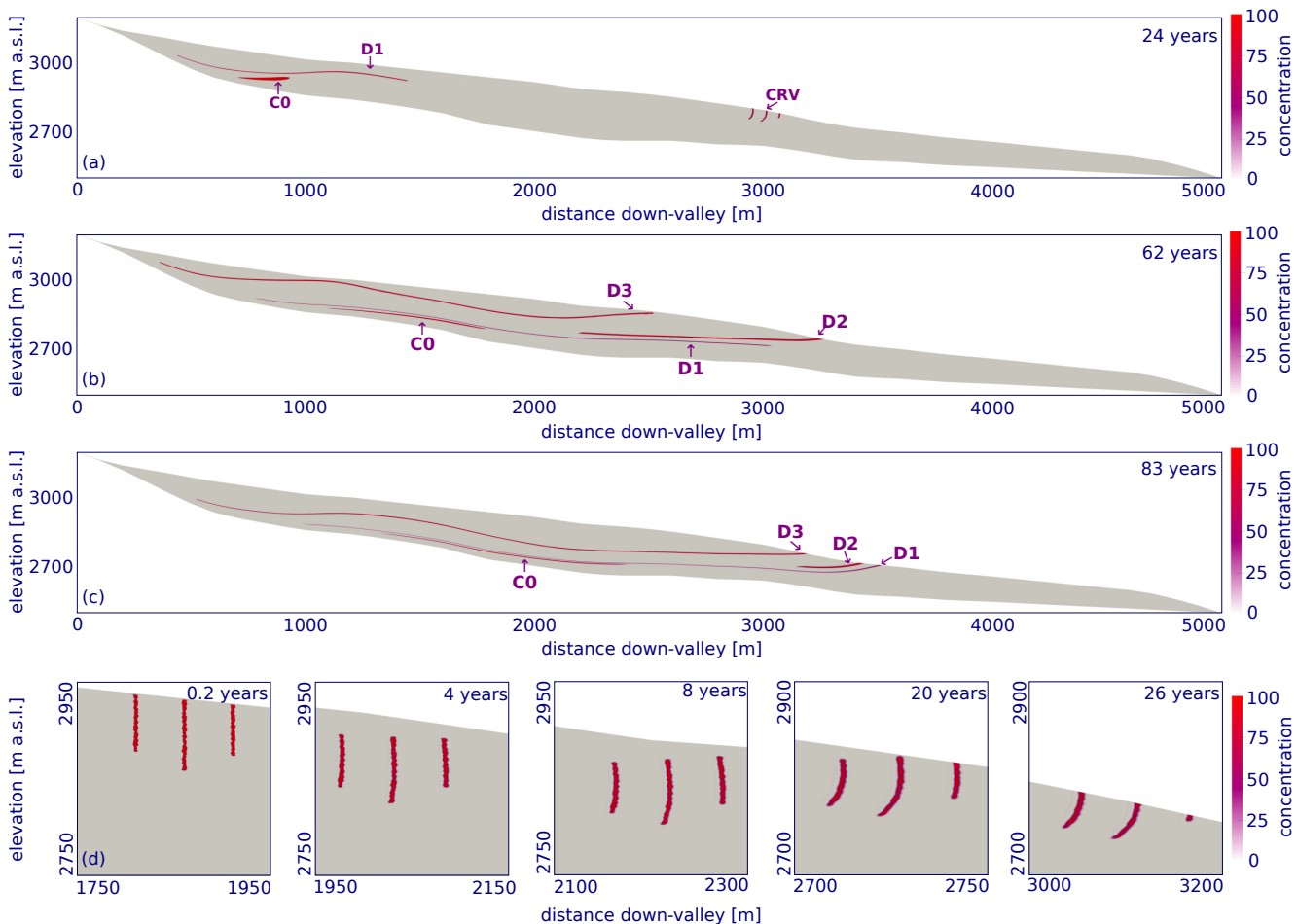

**Figure 9.** Results of the debris transport simulations for the 2D longprofile of Haute Glacier d'Arolla, where C0 indicates the circular debris inclusion, D1-D3 the surface debris layer deposits and CRV the crevasse-fills. (a) Debris concentration at 24 years, (b) at 62 years and (c) at 83 years after start of the simulations. Concentrations are displayed in the range of 0 to 100, numerical oscillations as excursions beyond the initial values of 0 or 100 are of magnitude less than $\pm 17$ and are truncated to the data limits. (d) Zoom of the crevasse-fills at 0.2, 4, 8, 20 and 26 years after the start of the simulations.

## 5.3 Idealized 3D glacier test

The deformation of englacial features shown in 2D is also represented in the 3D cases. The initially spherical inclusion becomes severely elongated in downglacier direction, forming a 'comet-like' tail as it is transported through the glacier. In addition to the downglacier elongation, where the glacier becomes narrower, the orographically left side of the debris inclusion is tilted upwards, and the centre of concentration is displaced laterally due to unequal lateral compression in the glacier flow field as it rounds the bend in the idealized valley (Fig. 10).

## 6 Discussion

### 6.1 Model capabilities and applicability

The debris transport and deformation modelled here reproduces structures analogous to those observed in structural glaciology, where elongated, sometimes cross-cutting debris layers outcrop with a range of dip angles at the glacier surface (Jennings et al., 2014; Goodsell et al., 2005). Not only can these structures be reproduced, but the 2D glacier simulations indicate that these elongated, band-shaped debris layers can form from initially fundamentally different debris deposits. In these simulations, ash fall or avalanche events that uniformly cover wider portions of the accumulation area are included as layer-shaped debris deposits at the glacier surface. Rockfall events that result in a locally thick debris deposit are represented by a circular inclusion, as an end-member case of possible remnants thereof. Both distinctly different debris inputs become severely elongated and band-like shaped during transport. The degree of elongation depends on the input location and, hence, the trajectory through the glacier. In addition to horizontal stretching, the effect of lateral compressional flow is shown in the 3D glacier simulations.

The work of Kirkbride and Deline (2013) illustrates how both the thickness and angle of emergence of a debris band play a role in determining the initial thickness of an emergent debris deposit. This, in combination with the location of emergence is an essential prerequisite to predict the development and further evolution of debris cover. The simulations highlight that for spatially restricted debris deposition events, distinct debris bands form within the glacier that will lead to initially delimited areas of debris cover on the surface. Hence, an assumption of a uniform englacial debris distribution of constant englacial debris concentration (Naito et al., 2000) that would result in a continuously debris-covered glacier surface where surface ablation is occurring, might not reflect reality adequately in order to capture the geometrical response of the glacier to the developing debris cover. The model presented here allows us to simulate the advection of debris concentration through a glacier in great detail. By explicitly modelling changes of debris concentration distribution as a result of transport within the glacier, local concentration changes (Eulerian perspective) caused by the deformation of debris deposit shape (Lagrangian perspective, cf. Fig. 10) during englacial transport can be captured. Hence, for a given debris deposition event and glacier geometry, we can quantify the exact amount of debris concentration at any point in space and time. As a result, we can track the englacial debris transport and quantify the timing, location and debris concentration of a debris band emerging in the ablation zone, as well as quantifying how the location of maximum debris emergence from a debris band, and its dip angle, will change over time. This is all critical information for determining how the spatial pattern of surface debris thickness will develop and evolve in time.

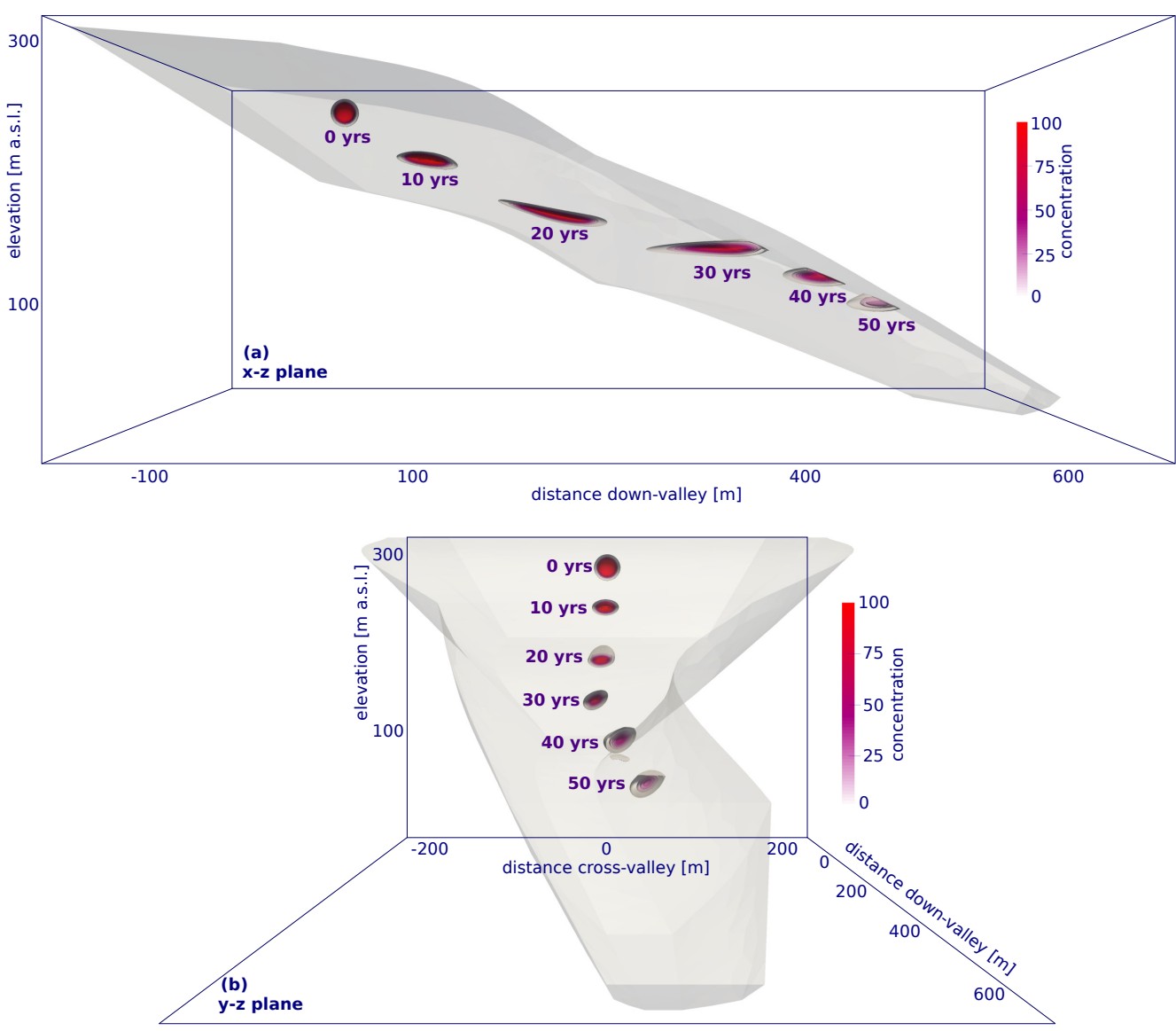

**Figure 10.** Results of the debris transport simulations for the 3D idealized glacier. Debris concentration isosurfaces of concentration 1, 10-100 in steps of 10 are displayed for a cut through (a) the x-z plane and (b) the y-z plane, as labelled in the figure. The isosurfaces shown refer to time $0-50$ years after the start of the simulation in an interval of 10 years. Concentrations are displayed in the range of 0 to 100, numerical oscillations as excursions beyond the initial values of 0 or 100 are of magnitude less than $\pm 4$ and are truncated to the data limits.

These results are also important in the context of the response of debris-covered glaciers to changes in climatic forcing or debris supply. Debris-covered glaciers are known to show distinctly different behaviour to clean-ice glaciers. This is due to the impact of debris cover on ice melt (Östrem, 1959; Mattson et al., 1993), which mainly depends on its thickness (Nicholson

and Benn, 2006; Reid and Brock, 2010). In the case of negative mass balance-conditions, the emergence of thin debris cover at the upper end of the ablation zone can lead to locally enhanced melting, lower the surface slope and alter the dynamic regime of the glacier (Benn et al., 2012). When and where those transitions occur is also related to the location and rate of debris emergence.

The model presented here, which resolves the governing physical processes without parameterization, and is based on a comprehensive numerical framework, offers a powerful tool with which to examine the validity of assumptions made in simpler models. For example, this model can be used to explore how the manner of prescribing debris (localized or distributed, spatially variable or constant, frequent or rare) affects the manner in, and timescales over, which a surface debris cover develops. This is valuable in the study of poorly understood earth systems like debris-covered glaciers, which evolve over timescales too long

to allow real world observations to answer these questions. The model presented here can be used to track the passage of any material through the glacier - under the assumption that the transported material itself is not significantly altering the glacier flow field. It therefore has potential applications not only for understanding the development of supraglacial debris layers, but also for interpreting observed structures in glaciers related to specific tephra deposits or rockfall events, for example. This model also offers the possibility to test the findings of studies that use patterns of englacial debris distribution on Antarctic

debris-covered glaciers to infer climate information at orbitally-paced time scales (Mackay and Marchant, 2017).

## 6.2    Model performance and limitations

The numerical accuracy of the presented model is dictated by the refinement and stability thresholds selected and will also vary dependent on the dimensions of the debris inputs. Nevertheless, we have demonstrated that the model performs satisfactorily in comparison to benchmark standards in the literature, and that this performance quality applies to the given model set up and

thresholds used to simulate the glacier cases presented.

The ratio of debris input size *versus* the total size of the glacier requires very fine mesh sizes to actually resolve the debris inputs, their transport and associated deformation. For example, for the presented model set up in the 3D glacier example, the total number of cells is in the order of $10^8$. This leads to high computational costs and the available computing resources impose constraints on the size of debris inputs the model is able to adequately simulate. In the case of localized debris inputs,

our mesh refinement approach has the potential to reduce the total number of cells substantially compared to a mesh that is globally refined. In the simulations presented here, the parameters in the refinement module are chosen to produce a mesh that is as coarse as possible while guaranteeing mass conservation (> 99 %), numerical stability and limiting numerical oscillations and numerical smearing to the levels presented in the Results (Sect. 5). Although numerical instabilities such as non-physical spurious oscillations and numerical diffusion are reduced efficiently by the approaches described in Sect. 3, numerical smearing

cannot be eliminated completely. It's magnitude is controlled by the mesh resolution (see Sect. 1). Decreasing cell size reduces numerical diffusion, but limitations of computing power will in practice impose a lower bound on cell size. Therefore, the magnitude of numerical diffusion expected for a given model set-up should be taken into account when interpreting model results. The 2D benchmark tests show that the chosen cell area threshold of $0.075 \ \mathrm{m}^2$ (for the 2D simulations) effectively limits non-physical spurious oscillations and numerical smearing below the levels presented in Sect. 5.1.

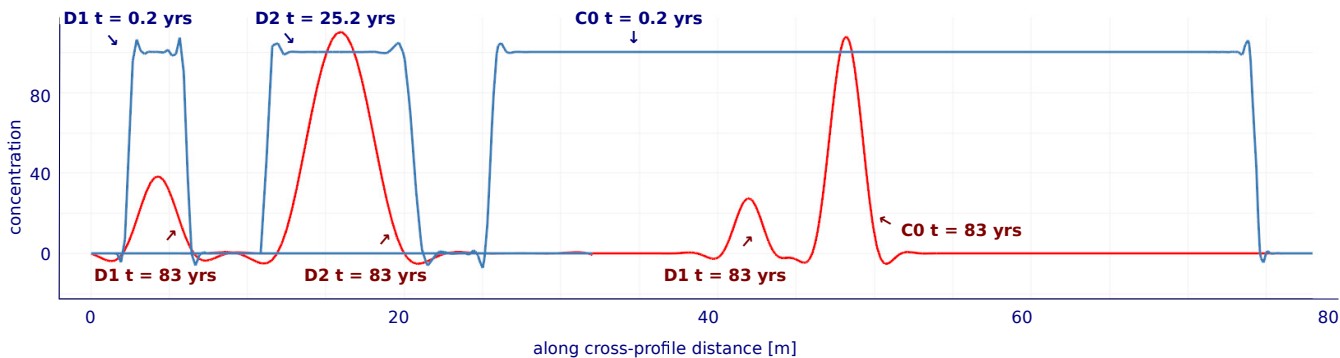

**Figure 11.** Vertical cross-sections at different model times for debris features D1, D2 and CO from the 2D glacier test (cf. Fig. 9). The vertical cross-sections are taken at the initial feature geometries in blue and evolved feature geometries at the simulation end in red. Corresponding cross-sections are centered at the same along cross-profile distance for ease of comparison. The vertical cross profiles are taken at a horizontal distance of 601.5 m (D1 $t = 0.2$ years) and 3148.5 m (D1 $t = 83$ years), 1100 m (D2 $t = 25.2$ years) and 3270 m (D2 $t = 83$ years), 502 m (C0 $t = 0.2$ years) and 2062 m (C0 $t = 83$ years). The exact location and orientation of the profiles is indicated in Fig. A8 in the supplementary material.

To demonstrate the concentration changes during transport within debris features for the 2D glacier test and its relation to numerical diffusion controlled by the cell area threshold (i.e. mesh refinement parameter), we plot concentration cross-sections for different debris features and model times in Fig. 11. The preservation of a central concentration peak over time is a good measure for identifying minimal numerical diffusion cases, as the redistribution of concentration by numerical diffusion would instantly lower that peak. In contrast, redistribution of debris concentration by transport preserves that peak due to the incompressibility of ice (cf. Sect. 1). Debris feature D1 experiences a strong amount of numerical diffusion over its $82.8$ years of transport through the glacier and its central concentration decreases by $\sim 62$ %. This performance deficit is due to the choice of a cell area threshold being too large to preserve this initially thin debris feature. In contrast to D1, the central concentrations of debris features D2 and C0 are clearly well preserved (maximum concentration values above $100$) and their change in concentration distribution (thinning of the cross-sectional length) is mostly due to transport. As a measure for debris band thickness over time we take full width at half maximum (fwhm) of the plotted debris concentration cross-sections in Fig. 11. D2 decreases from an initial fwhm of $9.3$ m to a fwhm of $4.4$ m over $57.8$ years of transport. C0 decreases from an initial fwhm of $48.8$ m to a fwhm of $2.2$ m over $82.8$ years of transport. Both cases clearly demonstrate the debris concentration changes caused by the transport through the glacier with minimal numerical diffusion. This highlights that a suitable cell area threshold choice is paramount to correctly simulate debris transport, as demonstrated with features D2 and C0. To improve the simulation results for feature D1, the only requirement is to lower the cell area threshold to a suitable value.

In 3D, the constraints on cell size are even more restrictive in terms of numerical stability and numerical diffusion. However, increase in computational costs in 3D is non-linear. The results we show here in the case of an idealized 3D glacier geometry will be subject to some numerical smearing, however the 3D benchmark test shows that numerical smearing can be minimized

by choosing a suitable mesh refinement parameter. Higher accuracy for representing sharp concentration variations can easily be achieved by changing mesh refinement cell size variables at increased computational cost.

In practice, application details dictate the constraints on model accuracy required to be able to adequately resolve the problem at hand. In this respect, by performing multiple simulations, the model can also be used to quantify the smearing of concentration features that arises from the choice of a coarser, but computationally feasible, cell size. Thus the introduced error by coarse mesh size choices can be easily quantified.

## 7 Conclusions and outlook

We developed a model to simulate debris transport within glaciers based on an advection algorithm that is coupled to a full-Stokes ice flow model. To facilitate computations and provide the spatial resolution required to accurately represent observed debris inputs, a localized mesh refinement strategy is employed. In this manner, the deformation of debris inputs, arising from gradients in the glacier's velocity field, can be modelled explicitly. This is crucial, as the location of emergence as well as amount and rate of debris emergence on the glacier surface, depend on the deposition location of debris inputs and are subsequently controlled by englacial transport and deformation. This is the first model capable of resolving transport and deformation of debris inputs in this detail. The advection algorithm combined with the full-Stokes approach offers the potential to model englacial transport of various debris inputs and for complex glacier geometries. In a future step, coupling the englacial transport model presented here to a (i) debris-aware surface mass-balance scheme and (ii) supraglacial debris transport scheme will enable us to fully model the co-evolution of debris cover and glacier geometry, and the behaviour of debris-covered glacier systems in general. Additionally, this will offer a powerful means by which to evaluate simpler representations of debris cover development within glacier systems.

## 8 Code availability

A development version of the source code of the core components of the model and a test example of Sect. 4 is available under the GNU General Public License V3 and can be found at https://github.com/awirbel.

*Acknowledgements.* This work was funded by the Austrian Science Fund (FWF), projects: P28521 and V309. Thanks to Christoph Mayer for comments on an early draft of this paper.

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
