# Peer review of "Modelling debris transport within glaciers by advection in a full-Stokes ice flow model"

_The Cryosphere, 2017_

## Referee Comment (RC2) · Anonymous Referee #2 · 21 Jul 2017

**Review of Wirbel et. al 2017 manuscript submitted to The Cryosphere**

In this contribution, Wirbel et. al. develop and test a new 3D physics-based model for the transport of englacial debris. The authors build off of an existing full-Stokes model for ice velocities, but (1) develop a new model for the debris advection component and (2) spend a considerable amount of effort developing and testing a scheme for efficient mesh optimization. The resulting model is evaluated against benchmark tests and a few idealized 2-D and 3-D englacial debris transport scenarios.

This paper is an excellent example of thorough attention to detail in model development. The authors should be commended. Furthermore, the paper is well written,

well organized, and displays a solid command of the numerical challenges involved in addressing their intended problem. From a technical and execution perspective, I have very little criticism. This work will make a significant contribution to an important and increasingly topical aspect of mountain glaciology. I very much look forward to future developments of the model (coupled ablation / mass-balance routines based on surface debris, surface transport, non-steady state evolving glacier geometries, etc.).

**General Comments**

My only general concern is related to the fit of this paper to the readers of The Cryosphere. Apart from the introduction, and the (understandably) preliminary 2-D and 3-D steady state glacier tests this paper contains very little that is relevant to the more general glaciology audience. The vast majority of the manuscript is concerned with the numerical model development, optimization, and benchmark testing. I am not criticizing the paper and think that it should be essentially published as is, but I will leave it to the editor to decide if it may be more appropriate for a different journal.

It would be a significant change, and I suggest this as being purely optional, but you may consider moving more of the benchmark results (and the detailed discussion thereof into the supplementary documents). This would serve to focus the paper more on the glaciological applications of the model. I have also made some suggestions below on where you can add more references to relevant field work that shows these types of englacial debris features. Including these may also help broaden this paper and bring it back to a more general glaciology audience.

You go to great lengths to model the change in the concentration of the debris deposit as it is advected. However, all that can be seen from your figures of the 2-D test are the modelled changes in the geometry of the debris deposit. This is too bad, because,

(as you say in P3, L7), the basic location and hence changing geometry of the overall debris deposit could be modelled using simple streamlines. What you bring to the table is much more powerful, however. If possible, I suggest changing the color bar / color scheme on the panels of Figure 9 so that the change in concentration as the material is advocated can actually be seen. However, I realize this may be impossible now.

**Specific comments**

P1, L18: This is still true for debris covered glacier systems that undergo no melt (cold-based alpine glaciers in Antarctica, for example. See [Kowalewski et al., 2011 ] or [Mackay and Marchant, 2016 ]). Change "melt " to "ablation "

P1, L18: I suggest that you change "... *and* transport of rock... " to "...*or* transport of rock... " Although it is unusual to find these decoupled, there could be situations, where debris supply is high and yet, due to a significant slope, the transport is so efficient that you never develop a large debris-covered ablation zone.

P2, L2-7: Several other authors have suggested / developed this idea as well. You may want to include some additional references. i.e. [Ackert, 1998 ; Clark et al., 1998 ; Monnier and Kinnard, 2015 ; Shroder et al., 2000 ] and others.

P2, L9: Also look at the work of [Reznichenko et al., 2011 ]

P3, L11: Other field studies have shown or inferred this as well. Look at the work of [Mackay et al., 2014 ]

P3, L18: This is a reasonable assumption for this iteration of the model. However, I hope that future models versions may be able to assign spatially heterogeneous rheological properties based on debris concentration.

P5, L23. P6, L1: It is not required, but I suggest you consider moving these sections (3.1 and 3.2) from the main text and putting this information into the supplemental documents. You are mostly describing tools and models that are already published.

Unless you have modified them, then you don't really need to describe them here again.

P6, L33. This adaptive mesh refinement is excellent. Do you also coarsen the mesh behind (upstream) of the deposit once it has transitioned down the streamline?

P7, L11. You did not show results of this "comparison " in this manuscript correct? I think that you mean that in general, the results are similar to those in Frutos et. al. 2014. - which is fine - but the way this sentence is written it sounds like you have actually done the comparison and included them in your results here. For clarity, I suggest that you change the sentence slightly to read: "Comparing *Results* of the benchmark test in Sec 5.1 derived with our approach *compare well* with adaptive mesh refinement based on a posteriori error estimation (de Frutos et al., 2014)*, and* demonstrate that. . ..."

P9, L21: Define *uh*, *ue* and *L2* in the equation

P9, L26: here you have defined *T* as total time. I do not think that that is what you are using for *T* in equations (1a) and P4, L18.

P10, L2: It is interesting that you choose a value for *A* that is best suited to temperate glaciers, but then use a no-slip boundary condition at the glacier –bed interface. I understand the no-slip boundary for these tests, but this condition is more consistent with cold-based polar glaciers which would have colder ice and a different value of *A*.

P10, L27: Remove sentence beginning: "Thereby, analysis of. . . " This sentence is unnecessary.

[Figure]

P11, L9: How big of a "bump" did you put in the subglacial topography? The reader has no way of knowing the characteristics of this or any other subsurface features based on the information and figures shown (see also my comment on Figure 5). Without this knowledge we cannot evaluate the impact (which I suspect is very little) that this should have on the simulation results.

P13, L7: You describe the results of the rotational flow test (Fig 6) in great detail, but then barely mention the swirling flow case (Figure 7). Is there a reason for this? In any case, I suggest that you move the swirling flow case (Fig. 7) to the Sup docs.

P15, L24-25: It is too bad that you did not show results from these simulations (using layer-shaped features). Although the sphere test is interesting, the layer deposits are more applicable to glaciological problems and questions. Perhaps these results were too difficult to visualize in 3D?

P17, L1-3: It would be helpful if you include a reference as examples of instances where this has been the assumption.

P17, L1-13: It is interesting in this discussion that you have not emphasized the importance of also of now being able to quantify the changes in the debris *concentration* as it moves and deforms down-glacier. I would mention this. Although out of scope of this study, determining the rate of debris cover formation in the ablation zone (which is one of the main potential applications of this model once it is linked to an ablation model) is directly linked to the debris concentration and thickness of the emerging debris bands. Your model allows this to now be predicted.

P17, L7-12: I'd would also recommend that you take a look at the work of [Mackay and Marchant, 2017 ] where englacial debris layers are directly linked to modelled changes

in the environmental conditions in the accumulation zone (at orbitally-paced time scales). Mentioning that being able to test theories like this and similar shows another area in which your model can be very useful and adding this into the discussion would broaden the perceived applicability of your work.

P18, L1-2: How hard would it be to implement a debris concentration - dependent rheology into your modelling framework? This would be excellent to have in future iterations of the model. See similar comment above.

**Technical Corrections**

P1, L4: Change "As debris is..." to "Because debris is..."

P1, L6: Change "...surface requires that the englacial transport pathways and deformation can be known." to "...surface requires knowledge of the englacial transport pathways and deformation."

P2. L13: Change "...get the full..." to "...model the full..."

P2. L17: Add comma "... Anderson, 2016), but as..."

P3. L25: Remove: " as incompressibility enforces conservation of ice density and hence ice volume." this clause is not necessary.

P3, L29: Start a new paragraph at the sentence "To solve the ..."

[Figure]

P3, L35: Consider deleting or moving the rest of this paragraph starting with the sentence: "In a later stage. . . " This information is better suited to the "Conclusions and outlook " section at the very end of the paper.

P4, L6, L7: Be consistent with using either "Section " or "Sec. "

P4, eqn. (1a) and P4, L18: define " $T$ " somewhere

P5, eqn. (5b): define $\partial\Omega_D$ . I assume that this is supposed to be $\partial\Omega_{bed}$ . If not, then unless $D$ is for the diffusion coefficient, choose another notation.

P6, L19: At sentence: "For 2D simulations. . . " Start new paragraph?

P6, L26: At sentence: "For 3D simulations. . . " Start new paragraph?

P6, L27: You have not yet introduced the refinement time step and thus this is confusing. I suggest ending the sentence with a reference to section 3.4. I.e.: ". . .at every refinement time step (see Sec. 3.4). "

P7, L2: This sentence is awkward and should be reworded.

P7, L15: Switch the order of sections 3.4 and 3.5. This improves readability and otherwise the reader does not know what SUPG is when you introduce it in P8, L3.

P9, L19: Reword the sentence starting with "Here, we. . . " It does not make sense as written.

P12, L9: Edit this sentence for better clarity. I suggest: "The results of benchmark tests 1 and 2 following the Bochev et. al (2004) are shown in . . . "

P12, L13: delete ", exemplary "

P12, L16: You refer to the case of *both* refinement time steps. However, so far you have only talked about the single time step (0.1 $\pi$) that is used in Figure 6. I know that you are also talking about the 0.01 $\pi$ time step (shown in the sup docs), but it is not clear the way it is written right now. Please edit for clarity.

P15, L22: change ". . .glacier is becoming narrower. . . " to ". . .glacier becomes narrower. . . "

P16, L1-3: This sentence is too long and confusing and needs to be edited. I suggest deleting the unnecessary extra qualifiers: ". . .that uniformly cover wider portions of the accumulation area. . . " and "resulting in thick debris deposit but limited in area. . . "

P16, L2: Change ". . .inclusion as a possible. . . " to ". . .inclusion representative of a possible. . . "

P18, L15: change start of line to . . . "*that is* as course as possible. . .

P18, L27: Start a new paragraph at this sentence.

**Figure Comments:**

Figure 1: Include a north arrow and scale bar

Figure 3:

- Please mark the ELA / beginning of the ablation zone. Although the reader can infer this from where the vertical velocity component passes zero from negative to positive, the addition of a simple arrow or line would be appreciated and aid in interpretation of Figure 9.

Figures 3 and 4:

- Since this geometry represents a glacier, but I suggest that you label x-axis and y-axis accordingly. i.e, "distance (m) " and "elevation (msl) "

Figure 5:

- I'm not sure why you have rotated the view individual panels unless you are trying to show the overall geometry. If this is the case, then it is not effective but rather just makes the figure look rather messy and less clear. It is not required, but if possible, I suggest that you show all output in the same orientation.

- Rather than using the axis labels x-axis, y-axis z-axis, consider using the physical interpretations for the labels (i.e. elevation, distance down-valley, distance cross-valley)

- A wireframe showing just the glacial bed would be appreciated. As it is now, the reader has no way of knowing what the subglacial topography looks like or where the bedrock "hump " is located.

- This is out of main scope of the debris transport focus of the paper, but I am curious as to why there is such a pronounced positive vertical velocity component at the upper right side of the glacier near the valley bend. There is compressive strain encouraging the emergence of ice here, but the magnitude surprises me.

Figure 6:

- The figure is fine. The caption could use some style adjustments for readability.

- In this and all figure captions where you have separate panels, you do not need to use the phrase: "In (a) etc…..is shown " Just start with the intended panel letter and say what it is as a separate sentence.

- Remove the sentence: "The data ranges from -0.14 to 1.11. " We can see that from the figures.

- Move the sentence "Color scales show concertation values " to the beginning or end of the caption.

- Shorten the final sentence to: (f) Results of the convergence test as a function of mesh refinement parameter *cvol*.

Figure 7 caption:

- Same style change comments as for figure 6.

Figure 8:

- The first two panels fail to convey the intended information. I think that there are two separate surfaces represented for each contour (except 90) when I zoom way in, but it is barely possible to resolve these as separate at the print level of figure zoom. Also, which surface is the FEM solution and which is the analytical? These are the same color and are not labeled or marked. I recognize that they are basically the same – which may be the point of the figure, but as it is now, it is just unclear. I suggest removing the panels (a) and (b) and just leaving in panel (c) which does convey usable information.

Figure 9:

- The color scale (red gradient) for concentration is not effective at conveying any information as all the lines basically look like the same shade of red. It may be impossible now, but if possible, consider changing this to a multiple-color colors scheme so that the change in concentration can actually be seen.

- Panels (a-c): Label the x-axis ('distance down-valley (m)). You probably only need to do this once for all panels.

- Label (draw an arrow or something) the various debris layers (D1, D2, D3, etc.). Right now, it is not possible to track which is which layer.

- Why not label the actual panel somewhere with their simulation times (24 yrs, 62 yrs, 85 yrs )

- Panel (d): label the x and y-axis correctly and put in at least two number values on each of the axes. Otherwise, the reader has no idea of the spatial scale.

- Why not label the actual panels with their simulation times (0.2 yrs, 8, yrs, 20 yrs, 26 yrs)

- Caption: Remove unnecessary sentence: "Concentrations are displayed in the range of 0 to 100. " You already show this on the color scale.

Figure 10:

- A rough 2-d outline of the glacier (surface, bedrock along the centerline in the x-z plane and the glacier sides in the x-y plane) would be helpful for interpretation. This does not have to be exact.

Please label the axis relative to the glacier model (i.e. elevation, distance down-valley, distance cross-valley)

- Label the debris distributions with their simulation times directly on the figure. There is plenty of room and this would make interpretation easier.

-Caption: Remove unnecessary sentence: "Concentrations are displayed in the range of 0 to 100. " You already show this on the color scale.

**Supplementary Material comments**

This may be a problem with my video player (I tried several) or my download, but I cannot play some of the .avi movie files. Please check that these are not damaged.

These play successfully:

3d _benchmark1.avi

2d _glaciercase.avi

These do not load (error):

2d _benchmark _exp1t20.avi

2d _benchmark _exp1t200.avi

2d _benchmark _exp2t15.avi

2d _benchmark _exp2t150.avi

Figure A1:

Please put a scale or tick marks on the x-axis and y-axis. the reader ha no idea what the scale is. Or is this dimensionless?

Label the color bar in panels (a) and (b).

In panel (b), if all the velocities are the same, then state that in the caption. Putting the "1.22 " beneath the panel looks strange and does not convey any useful information.

Figure A2 and A3:

Label the panels in the left hand column (2D concentrations) with the dt used in that row. The left hand panels need x-axis and y-axis labels/tick marks (0 – 1?), otherwise the reader has no way of knowing where the profiles in the middle and right hand panels are taken from. You may also want to put two dashed lines across the concentration panels that show the location of the profiles.

It is odd that you put your *References* section in the middle of the document before Figure A2. Maybe this is something that happened in the auto-collate process during submission? I suggest just moving all Sup Doc references to the end of the document.

REFERENCES

Ackert, R. (1998), A rock glacier/debris-covered glacier system at Galena Creek, Absaroka Mountains, Wyoming, *Geografiska Annaler. Series A. Physical Geography*, *80*, 267-276, doi:10.1111/j.0435-3676.1998.00042.x.

Clark, D., E. Steig, P. N, and A. Gillespie (1998), Genetic variability of rock glaciers, *Geografiska Annaler. Series A. Physical Geography*, 175-182, doi:10.1111/j.0435-3676.1998.00035.x.

Kowalewski, D. E., D. R. Marchant, K. M. Swanger, and J. W. Head (2011), Modeling vapor diffusion within cold and dry supraglacial tills of Antarctica: Implications for the preservation of ancient ice, *Geomorphology*, *126*(1-2), 159-173, doi:10.1016/j.geomorph.2010.11.001.

Mackay, S. L., and D. R. Marchant (2016), Dating buried glacier ice using cosmogenic 3He in surface clasts: theory and application to Mullins Glacier, Antarctica, *Quaternary Science Reviews*, *140*, 75-100, doi:http://dx.doi.org/10.1016/j.quascirev.2016.03.013 .

Mackay, S. L., and D. R. Marchant (2017), Obliquity-paced climate change recorded in Antarctic debris-covered glaciers, *Nature Communications*, *8*, 14194.

Mackay, S. L., D. R. Marchant, J. L. Lamp, and J. W. Head (2014), Cold-based debris-covered glaciers: Evaluating their potential as climate archives through studies of ground-penetrating radar and surface morphology, *Journal of Geophysical Research: Earth Surface*, *119*(11), 2505-2540.

Monnier, S., and C. Kinnard (2015), Reconsidering the glacier to rock glacier transformation problem: New insights from the central Andes of Chile, *Geomorphology*, *238*, 47-55.

Reznichenko, N. V., T. R. Davies, and D. J. Alexander (2011), Effects of rock avalanches on glacier behaviour and moraine formation, *Geomorphology*, *132*(3-4),

doi:10.1016/j.geomorph.2011.05.019.

Shroder, J., M. Bishop, L. Copland, and V. Sloan (2000), Debris-covered glaciers and rock glaciers in the Nanga Parbat Himalaya, Pakistan, *Geografiska Annaler. Series A, Physical Geography*, *82*(1), 17-31, doi:10.1111/j.0435-3676.2000.00108.x.

---

## Author Comment (AC1) · 8 Sep 2017

Anna Wirbel1, Alexander Helmut Jarosch2, and Lindsey Nicholson1

1Institute of Atmospheric and Cryospheric Sciences, University of Innsbruck, Innsbruck, Austria 2Institute of Earth Sciences, University of Iceland, Reykjavík, Iceland

We would like to thank Garry Clarke for his thorough and valuable review and for providing helpful comments on our manuscript.

**1** Specific Comments**

- **Comment:** For me the main source of confusion was whether a dimensional or dimensionless treatment was being followed. 5 I got the impression that in fact both points of view were being taken but that the dividing lines were unclear. For example, the debris diffusivity D has dimensions  $m^{-2} s^1$  (as it appears in Equation 5a) but on Page 9, Line 14  $D = 10^{-6}$  suggests it has become dimensionless for the LeVeque test. Contributing to this confusion is the fact that time steps of  $0.01 \pi$  and  $0.1 \pi$  (dimensionless?) and mesh sizes of  $L_{csize} = 0.15$  m are discussed on the same page. Please clarify here and elsewhere.
- **Response:** Thanks for this comment, we were not completely clear about this point. We follow a dimensional treatment 10 throughout this study. In the case of the "rotating three body problem" (de Frutos et al., 2014) (Sect. 4.1), we enlarge the computational domain from  $\Omega = (0,1) \times (0,1)$  to  $\Omega = (0,100) \times (0,100)$  meters in order to allow direct evaluation of the performance of chosen mesh refinement thresholds, which have units, that we apply in the glacier cases. In this manner, the size of concentration features in the test case becomes comparable to that of the concentration features in the glacier cases. This allows us to evaluate the suitability of chosen mesh refinement thresholds ( $c_{vol}$  and  $L_{csize}$ ) that we also apply in the
- 15 presented glaciological applications. To facilitate the understanding of dimensions used, as well as to highlight the actual equation we solve in our model, we have added Eq. 1 (Equation 6a in the revised manuscript) to the manuscript. Assuming an incompressible fluid and a constant diffusivity (D), Eq. 5a becomes:

$$\frac{\partial c}{\partial t} = D\nabla^2 c - \boldsymbol{u} \cdot \nabla c. \tag{1}$$

The assumptions used here have been stated in the manuscript. From Eq. 1 above it becomes apparent that D has the dimensions of m2 s-1 when c is assumed to be dimensionless, as it is in our study. Timesteps as multiples of  $\pi$  in the advection tests stem from the angular velocity chosen in the de Frutos tests (de Frutos et al., 2014), where a whole rotation takes  $2\pi$  seconds. We have clarified the dimensional approach we take in the manuscript by adding information on the dimensions of parameters at the following locations in the manuscript:

- $D = 10^{-6} \text{ m}^2 \text{ s}^{-1}$ , Page 9 Line 14
- $\Omega = (0, 100) \times (0, 100)$  meters, Page 9 Line 15
- $0.01 \pi$  s;  $0.1 \pi$  s, Page 9 Line 19
- $2\pi s(T)$ , Page 9 Line 20
- 5 T, at t = 1.5 s, Page 9 Line 26
  - $2\pi s(T)$ , Page 9 Line 30
  - $\Omega = (0, 32) \times (0, 32) \times (0, 40)$  meters, Page 9 Line 31
  - $0.04 \,\pi$  s, Page 9 Line 32
  - $0.1 \pi$  s, Page 12 Line 14

**10 – $2\pi$ s, Page 12 Line 15**

- $0.01 \pi$  s;  $0.1 \pi$  s, Page 13 Line 4 and captions of Figs. 6-7
- dt = 0.1 s, dt = 0.01 s, dt = 0.001 s, dt = 0.0005 s, t = 0.5 s, y = 0.6 m, x = 0.75 m, y = 0.85 m, x = 1.0 m, Page 2 Lines 1-4, Lines 10-11, Line 13 and captions of Figs. A2-A5 in the supplementary material

**2 Technical Corrections**

15 We thank Garry Clarke for the careful reading of the manuscript and adapted the text to meet the suggested Technical Corrections.

**Comment:** *Figure 1 The caption should read "Kennicott Glacier" (spelling)* **Response:** Corrected.

**20**

**Comment:** *P03, L29 "Eulerian" (not Eularian)* **Response:** Corrected.

**Comment:** *P10*, *L04* 917 kg m-3

25 **Response:** Inserted space.

**Comment:** *P10, L30* x = 1800 m and x = 1900 m**Response:** Inserted units.

**Comment: P20, L19 Bozhinskiy et al. (1986): Check caps style**

**Response:** We checked the reference Bozhinskiy et al. (1986) and adapted the capitalization of the title according to the reference style used in The Cryosphere.

5

**Comment: P20, L28 Glen (1955): Check caps style**

**Response:** We checked the reference Glen (1955) and adapted the capitalization of the title according to the reference style used in The Cryosphere.

**10 **Comment:** *P21, L07 John and Novo (2011): Check caps style**

**Response:** We checked the reference John and Novo (2011) and adapted the capitalization of the title according to the reference style used in The Cryosphere.

**Comment: P21, L28 LeVeque (1996). Caps style**

15 **Response:** We checked the reference LeVeque (1996) and adapted the capitalization of the title according to the reference style used in The Cryosphere.

**Comment: P22, L15 Nye (1957): Caps style**

**Response:** We checked the reference Nye (1957) and adapted the capitalization of the title according to the reference style used in The Cryosphere.

**Comment: P22, L35 Ostrem (1959): Caps style**

**Response:** We checked the reference Östrem (1959) and adapted the capitalization of the title according to the reference style used in The Cryosphere.

25

**References**

Bozhinskiy, A., Krass, M., and Popovnin, V.: Role of debris cover in the thermal physics of glaciers, J. Glaciol., 32, 255–266, 1986.

- de Frutos, J., García-Archilla, B., John, V., and Novo, J.: An adaptive SUPG method for evolutionary convection-diffusion equations, Comput. Method. Appl. M., 273, 219–237, doi:10.1016/j.cma.2014.01.022, 2014.
- 5 Glen, J. W.: The creep of polycrystalline ice, Proceedings of the Royal Society of London A: Mathematical, Physical and Engineering Sciences, 228, 519–538, doi:10.1098/rspa.1955.0066, 1955.
  - John, V. and Novo, J.: Error analysis of the SUPG finite element discretization of evolutionary convection-diffusion-reaction equations, SIAM J. Numer. A., 49, 1149–1176, doi:10.1137/100789002, 2011.

LeVeque, R.: High-resolution conservative algorithms for advection in incompressible flow, SIAM J. Numer. A., 33, 627-665,

10 doi:10.1137/0733033, 1996.

Nye, J. F.: The distribution of stress and velocity in glaciers and ice-sheets, Proceedings of the Royal Society of London A: Mathematical, Physical and Engineering Sciences, 239, 113–133, doi:10.1098/rspa.1957.0026, 1957.

Östrem, G.: Ice melting under a thin layer of moraine, and the existence of ice cores in the moraine ridges, Geogr. Ann., 41, 228–230, 1959.

---

## Author Comment (AC2) · 8 Sep 2017

Anna Wirbel1, Alexander Helmut Jarosch2, and Lindsey Nicholson1

1Institute of Atmospheric and Cryospheric Sciences, University of Innsbruck, Innsbruck, Austria 2Institute of Earth Sciences, University of Iceland, Reykjavík, Iceland

We would like to thank Anonymous Referee #2 for detailed and helpful comments on our manuscript.

**1** General Comments**

**Comment:** It would be a significant change, and I suggest this as being purely optional, but you may consider moving more of the benchmark results (and the detailed discussion thereof into the supplementary documents). This would serve to focus

5 the paper more on the glaciological applications of the model. I have also made some suggestions below on where you can add more references to relevant field work that shows these types of englacial debris features. Including these may also help broaden this paper and bring it back to a more general glaciology audience.

**Response:** We decided not to change the overall structure of the manuscript, as it would take away the focus from the extensive benchmark testing, which we think is an important contribution of this manuscript. Still, we included additional field work

10 references, as suggested in the Specific Comments by Anonymous Referee #2.

**Comment:** You go to great lengths to model the change in the concentration of the debris deposit as it is advected. However, all that can be seen from your figures of the 2-D test are the modelled changes in the geometry of the debris deposit. This is too bad, because, (as you say in P3, L7), the basic location and hence changing geometry of the overall debris deposit

15 could be modelled using simple streamlines. What you bring to the table is much more powerful, however. If possible, I suggest changing the color bar / color scheme on the panels of Figure 9 so that the change in concentration as the material is advocated can actually be seen. However, I realize this may be impossible now.

**Response:** As a response to this comment, we want to revisit the meaning of changes in concentration in the model results. We present a model which simulates the evolution of the debris concentration field within a glacier. The changes in concentration

20 distribution are currently driven by advection with the glacier flow. As we treat glaciers as incompressible fluids, the resulting glacier flow fields are divergence-free and hence the concentration of debris in a control volume that is advected and deformed by the flow should not change (from a Lagrangian perspective). For a fixed volume in space, the concentration changes, as debris is advected through this fixed volume (from a Eulerian perspective) (see manuscript P3, L23-35). By modelling the concentration distribution, the model provides the required knowledge to model debris cover formation on the glacier surface

due to emergence of englacial debris bands. As a result of the incompressibility assumption, changes in concentration at the edges of the debris inclusions are purely a result of numerical diffusion and mesh resolution and can hence be taken as a key metric of model performance, which we show in the benchmark tests in Sect. 5.1.

- On P3, L7 we say that the location of an individual clast (single debris particle or the centre of an "undeformable" boulder) 5 can be calculated by simple streamline tracing. However, the exact location of emergence as well as the transient shape of a polymictic debris inclusion, which will be deformed when advected through a glacier, cannot be recovered by simple streamline calculations. Even more, this becomes exceedingly difficult when the glacier geometry is changing, and hence the velocity fields. In the Specific Comments, we address the comments on the changes in concentration and the use of the color scheme in Fig. 9 in more detail.
- 10

**2 Specific Comments**

**Comment:** *P1*, *L18*: *This is still true for debris covered glacier systems that undergo no melt (cold-based alpine glaciers in Antarctica, for example. See Kowalewski et al. (2011) or Mackay and Marchant (2016). Change "melt " to "ablation "*

15 **Response:** We changed "melt" to "ablation".

**Comment:** *P1*, *L18: I* suggest that you change "... and transport of rock..." to "... or transport of rock..." Although it is unusual to find these decoupled, there could be situations, where debris supply is high and yet, due to a significant slope, the transport is so efficient that you never develop a large debris-covered ablation zone.

- 20 **Response:** The sentence reads: "If debris supply and melting is sufficiently high, and transport of rock material out of the glacier system is inefficient, a debris-covered glacier can develop, where a large portion of the ablation zone is covered with a continuous layer of rock material (Kirkbride, 2011)." The "and transport of rock" implies that in any case, an inefficient transport of rock material out of the glacier system is required to be able to develop a debris-covered glacier, which is also true for the situation mentioned in the comment. Therefore, we chose to leave this sentence as it is.
- 25

30

**Comment:** *P2, L2-7: Several other authors have suggested / developed this idea as well. You may want to include some additional references. i.e. Ackert (1998); Clark et al. (1998); Monnier and Kinnard (2015); Shroder et al. (2000) and others.* **Response:** We now added citations for (Ackert, 1998; Clark et al., 1998) after the sentence that reads: "The implication of this process-continuum is that glaciers can transition between rockglaciers, debris-covered glaciers and clean ice glaciers through space or time as a result of the varying ice influx."

Comment: P2, L9: Also look at the work of Reznichenko et al. (2011)

Response: Thanks for this, we added a citation of the work of Reznichenko et al. (2011).

**Comment:** *P3, L11: Other field studies have shown or inferred this as well. Look at the work of Mackay et al. (2014)* **Response:** Thanks for this reference, we added a citation of the work of Mackay et al. (2014).

- 5 Comment: P3, L18: This is a reasonable assumption for this iteration of the model. However, I hope that future models versions may be able to assign spatially heterogeneous rheological properties based on debris concentration.
   Response: From a numerical point of view, it is rather easy to add spatially and temporally heterogeneous rheological properties. An excellent example is given by Aschwanden et al. (2012) with an entropy based rate factor A. Thus if a rheological parametrization based on debris content supported by convincing field data is published in the future, it can be incorporated in
- 10 the model framework. However, so far the knowledge on rheological properties of debris-laden ice is limited to a small number of shear tests (e.g. Fitzsimons et al., 2001) and a recent review of available observations and model approaches (Moore, 2014) highlights both the complexity of the rheology of ice-debris mixtures and the need for further laboratory or field testing of models.
- 15 **Comment:** *P5, L23. P6, L1: It is not required, but I suggest you consider moving these sections (3.1 and 3.2) from the main text and putting this information into the supplemental documents. You are mostly describing tools and models that are already published. Unless you have modified them, then you don't really need to describe them here again.*

**Response:** We thank Anonymous Referee #2 for this suggestion but chose to keep these sections in the manuscript to provide introductory information on the tools that are used to develop the presented model.

20

**Comment:** *P6, L33. This adaptive mesh refinement is excellent. Do you also coarsen the mesh behind (upstream) of the deposit once it has transitioned down the streamline?*

**Response:** Due to the way the mesh refinement is implemented, no mesh coarsening is needed upstream of the advected debris features once they have been transported further downstream. In every refinement time step, the refinement is performed on

- a mesh that is coarse over the entire domain and only the regions of interest i.e. where debris is present and its surroundings (whose extent depends on the chosen refinement time step and the actual velocity) are refined. As the same domain-wide coarse mesh is used for every refinement time step, the regions further upstream of the actual location of the debris feature, where it has been transported through previously, do not become refined in the subsequent refinement time step any more. This spares the need for mesh coarsening. This is expressed in the manuscript at the end of P6, Sect. 3.3.
- 30

35

**Comment:** P7, L11. You did not show results of this "comparison" in this manuscript correct? I think that you mean that in general, the results are similar to those in de Frutos et al. (2014). - which is fine - but the way this sentence is written it sounds like you have actually done the comparison and included them in your results here. For clarity, I suggest that you change the sentence slightly to read: "Comparing Results of the benchmark test in Sec 5.1 derived with our approach compare well with adaptive mesh refinement based on a posteriori error estimation (de Frutos et al., 2014), and demonstrate that ...".

3

**Response:** This is correct, thanks. We changed the sentence to: "The results of the benchmark test in Sec. 5.1 derived with our approach compare well with that derived using adaptive mesh refinement based on *a posteriori* error estimation (de Frutos et al., 2014), this demonstrates that ...".

**5 **Comment:** *P9, L21: Define uh, ue and L2 in the equation**

**Response:** We inserted the definitions by changing the relevant sentence to: "In order to evaluate the chosen cell area threshold, we perform convergence tests where (a)  $||c_{\rm h} - c_{\rm e}||_{\rm L_2} = \sqrt{\sum_{k=1}^{n} (c_{\rm h_k} - c_{\rm e_k})^2}$  the L2 norm of the error between the computed finite element solution  $c_{\rm h}$  and the exact solution  $c_{\rm e}$ , where n is the number of computation locations, and (b) the Root Mean Square (RMS) error between the computed finite element solution and the exact solution for different cell size thresholds are

10 computed. Therefore, we first compute the exact solution on the same mesh that is used in the finite element solution."

**Comment:** *P9, L26: here you have defined T as total time. I do not think that that is what you are using for T in equations (1a) and P4, L18.*

**Response:** In P4, L18 T is used to describe the transpose of the respective quantity. For clarity we altered it to  $\top$ . We also adjusted the text to use  $t_{\text{total}}$  instead of T to describe the total time.

**Comment:** *P10, L2: It is interesting that you choose a value for A that is best suited to temperate glaciers, but then use a no-slip boundary condition at the glacier-bed interface. I understand the no-slip boundary for these tests, but this condition is more consistent with cold-based polar glaciers which would have colder ice and a different value of A.*

- 20 Response: For these tests, the choice of the exact numerical value for A is not critical, as we were not trying to reproduce the actual debris transport of Haute Glacier d'Arolla, but to show the general features of debris transport within mountain glaciers. The choice of a smaller constant value of A, representative for cold-based glaciers and consistent with a "real world" no-slip boundary condition, would only cause a reduction of glacier flow velocities but not change the actual velocity patterns (see Fig. 1). This would increase the amount of time needed to transport the debris deposits downglacier, but would not induce a
- 25 change of the main features and characteristics of the transport and associated deformation of debris inclusions observed in the tests. In future applications of the model, where we are planning to simulate the evolution of specific debris-covered glaciers, we will apply site-specific basal conditions and appropriate flow law parameters representative for the thermal regime of the glacier. For clarity, we rephrased the first sentence of Sect. 4.2 to: "The purpose of these tests is to demonstrate the character-istics of debris transport within mountain glaciers, not to reproduce a particular event on a specific glacier. Hence, all velocity
- 30 computations are initialized with a no-slip condition at the glacier/bedrock boundary, the flow law exponent n is set to 3 and the Glen rate factor A is set to  $2.4 \times 10^{-24}$  s-1Pa-3, a standard value for temperate ice (Cuffey and Paterson, 2010)."

**Comment: P10, L27: Remove sentence beginning: "Thereby, analysis of. . . " This sentence is unnecessary.**

**Response:** This sentence aims to explain the chosen test setup, therefore we kept it in the revised manuscript but changed it to: "These debris deposits of varying size, shape and location of deposition were chosen to facilitate analysis of the interplay

Figure 1. Velocity of 2D glacier profile. (a) temperate conditions with  $A = 2.4 \times 10^{-24} \text{ s}^{-1} \text{Pa}^{-3}$ , standard value for temperate ice (Cuffey and Paterson, 2010) and (b) cold-based conditions with  $A = 3.5 \times 10^{-25} \text{ s}^{-1} \text{Pa}^{-3}$  a value for cold ice at roughly -10 degrees Celsius (Cuffey and Paterson, 2010)

between debris input location, deformation during transport and the zone of emergence."

**Comment:** *P11, L9: How big of a "bump " did you put in the subglacial topography? The reader has no way of knowing the characteristics of this or any other subsurface features based on the information and figures shown (see also my comment to the characteristics)*

5 on Figure 5). Without this knowledge we cannot evaluate the impact (which I suspect is very little) that this should have on the simulation results.

Response: We adjusted Fig. 5 to illustrate the subglacial topography of the idealized 3D glacier case.

Comment: P13, L7: You describe the results of the rotational flow test (Fig 6) in great detail, but then barely mention the
swirling flow case (Figure 7). Is there a reason for this? In any case, I suggest that you move the swirling flow case (Fig. 7) to the Sup docs.

**Response:** The swirling flow test is identical to the "rotating three body problem" but using a different velocity field, which forces the features to first change their shape but then recover their initial shape at total time t = 1.5 s. Hence, this offers an even more challenging test case for the advection module and we included the figure to demonstrate that our model is as well

15 capable of producing appropriate results in this test. Since the test cases are identical except the velocity field used, we keep the description short but we now changed the sentence on P13, L7 to: "Also, when the initial concentration pattern is subjected to a more complex, swirling flow (LeVeque, 1996), the results of these more challenging test simulations again show satisfactory model performance, as can be seen in Fig. 7 and Fig. A5 in the supplementary material."

**Comment:** *P15, L24-25: It is too bad that you did not show results from these simulations (using layer-shaped features). Although the sphere test is interesting, the layer deposits are more applicable to glaciological problems and questions. Per-haps these results were too difficult to visualize in 3D?*

**Response:** We are aware of the glaciological relevance of layer deposit simulations, however we chose to not present initial layer results at this stage as they deserve a detailed discussion and would be more informative if they were based on a real world example. We plan to perform such a study in the near future.

**Comment:** *P17, L1-3: It would be helpful if you include a reference as examples of instances where this has been the assumption.*

10 **Response:** We included a reference of the work of Naito et al. (2000), where this assumption has been made.

**Comment:** *P17*, *L1-13*: It is interesting in this discussion that you have not emphasized the importance of also of now being able to quantify the changes in the debris concentration as it moves and deforms down-glacier. I would mention this. Although out of scope of this study, determining the rate of debris cover formation in the ablation zone (which is one of the main potential applications of this model once it is linked to an ablation model) is directly linked to the debris concentration

15 main potential applications of this model once it is linked to an ablation model) is directly linked to the debris and thickness of the emerging debris bands. Your model allows this to now be predicted.

**Response:** The current study focuses on demonstrating a numerically stable advection scheme that can be used to predict transport and deformation of debris features, as Anonymous Referee #2 points out above. However when looking at "local" debris concentrations and their potential change over time, one has to be careful to describe the reference frame in which these

- 20 changes take place. From a fluid dynamics point of view, the concentration of debris in a control volume that is advected and deformed by the flow should not change when the fluid is incompressible. This is described in the manuscript on P3, L23-35. We currently focus on that property of the model as it demonstrates the performance of our numerical implementation when we assume D, the debris diffusivity, to be very small, or even zero, and thus advection being the dominant transport process. In contrast to this Lagrangian viewpoint, a more applied (even Eulerian) viewpoint would be to investigate a fixed volume in
- 25 space and time. In this case, debris concentration changes over time and such changes are very interesting for the interaction of debris with mass balance processes or the formation of emerging debris bands, as Anonymous Referee #2 points out above. We have added a sentence to the manuscript to highlight this potential application in future studies: "The model presented here allows us to simulate the advection of debris concentration through a glacier in great detail and therefore any resulting local concentration changes (Eulerian perspective), e.g. the deformation of debris deposit shape (Lagrangian perspective, cf. Fig. 10) "

30 10)."

35

**Comment:** *P17*, *L7-12*: *I'd* would also recommend that you take a look at the work of Mackay and Marchant (2017) where englacial debris layers are directly linked to modelled changes in the environmental conditions in the accumulation zone (at orbitally-paced time scales). Mentioning that being able to test theories like this and similar shows another area in which your model can be very useful and adding this into the discussion would broaden the perceived applicability of your work.

**Response:** Thanks for this reference. We included the following sentence at the end of Sect. 6.1: "This model also offers the possibility to test the findings of studies that use patterns of englacial debris distribution on Antarctic debris-covered glaciers to infer climate information at orbitally-paced time scales (Mackay and Marchant, 2017)."

5 Comment: P18, L1-2: How hard would it be to implement a debris concentration - dependent rheology into your modelling framework? This would be excellent to have in future iterations of the model. See similar comment above.
 Response: In principle, the implementation would be rather simple. Compare comment above.

**10 3 Technical Corrections**

We thank Anonymous Referee #2 for the detailed Technical Corrections suggested for the text of the manuscript.

**Comment:** *P1, L4: Change "As debris is..." to "Because debris is..."* **Response:** Done.

**15**

**Comment:** *P1, L6: Change ". . .surface requires that the englacial transport pathways and deformation can be known. " to ". . .surface requires knowledge of the englacial transport pathways and deformation. "* **Response:** Done.

20 **Comment:** *P2. L13: Change "...get the full..." to "...model the full..."* **Response:** Done.

Comment: P2. L17: Add comma "... Anderson, 2016), but as... " Response: Done.

**25**

**Comment:** *P3. L25: Remove:* " as incompressibility enforces conservation of ice density and hence ice volume." this clause is not necessary.

**Response:** We now rewrite the sentence as follows: "Assuming that ice is an incompressible fluid, and consequently that the ice flow fields must be divergence-free, any deformational patterns inducing horizontal elongation, must, at the same time, cause

30 vertical compression. In the context of englacial debris transport, this implies that the initial debris concentration is constant for an initial control volume of ice being tracked (i.e. seen from Lagrangian perspective)."

Comment: P3, L29: Start a new paragraph at the sentence "To solve the ... "

**Comment:** *P3*, *L35*: Consider deleting or moving the rest of this paragraph starting with the sentence: "In a later stage. . " This information is better suited to the "Conclusions and outlook" section at the very end of the paper.

- 5 Response: We included this sentence in the Introduction to clarify the scope of this paper, but also the context in which we are developing this model. We changed the sentence to: "The model presented here forms part of an envisaged fully-integrated model framework that, by including (1) a free-surface evolution scheme including debris-aware mass balance subroutines and (2) a transport model for debris at the glacier surface interacting with the mass balance subroutines, will be capable of simulating the transient response of debris-covered glaciers, with predetermined debris inputs, to a changing climate."
- 10

**Comment: P4, L6, L7: Be consistent with using either "Section" or "Sec."**

Response: We replaced "Section" and "Sec." with "Sect." following the guidelines for manuscript preparation of The Cryosphere.

**Comment: P4, eqn. (1a) and P4, L18: define "T" somewhere**

15 **Response:** We have changed "T" to the symbol " $\top$ " as a more appropriate symbol to express that a quantity has been transposed.

**Comment:** *P5, eqn. (5b): define*  $\partial \Omega_D$  *. I assume that this is supposed to be*  $\partial \Omega_{bed}$  *. If not, then unless D is for the diffusion coefficient, choose another notation.*

**Response:** We changed  $\partial \Omega_D$  to  $\partial \Omega_0$  as in this case it is used to describe the boundary of the entire domain, except where input 20 locations are prescribed. We also included the following sentence in Sect. 4.2 to describe what boundary conditions are used in the glacier tests in this study: "In the presented glacier simulations, all debris inclusions have been deposited in a single event, hence they are all initialized as inclusions within the glacier, i.e. the entire glacier/atmosphere boundary belongs to  $\Omega_0$ ."

**Comment:** P6, L19: At sentence: "For 2D simulations. . . " Start new paragraph?

- 25 Comment: P6, L26: At sentence: "For 3D simulations..." Start new paragraph? Response: We start a new paragraph at the sentence: "For 2D simulations..", but as some of the concluding sentences correspond to both, the 2D and 3D implementations of mesh refinement, we keep the detailed descriptions of the 2D and 3D case in one paragraph.
- Comment: P6, L27: You have not yet introduced the refinement time step and thus this is confusing. I suggest ending the sentence with a reference to section 3.4. I.e.: ". . . at every refinement time step (see Sec. 3.4). "
   Response: We added the reference to Sect. 3.5 (as we switched Sect. 3.4 and 3.5) at P6, L18 where we first introduce the refinement time step.
- 35 **Comment:** *P7, L2: This sentence is awkward and should be reworded.*

**Response:** We changed the sentence to: "Adaptive mesh refinement strategies often employ *a posteriori* error estimation (e.g. John, 2000). The PDE is solved and the assigned error estimators and indicators are used to mark the cells for refinement and potentially coarsening. Subsequently, the marked cells become modified and the PDE is solved on the newly refined mesh. This process is repeated until the error estimators and indicators fall below a user-defined tolerance within every cell."

5

**Comment:** *P7*, *L15: Switch the order of sections 3.4 and 3.5. This improves readability and otherwise the reader does not know what SUPG is when you introduce it in P8, L3.* **Response:** Done.

10 **Comment:** *P9, L19: Reword the sentence starting with "Here, we. . . " It does not make sense as written.* **Response:** We changed the sentence to: "Here, we present results of computations using two different refinement time steps, (a) small refinement time step of  $0.01 \pi$  s and (b) a larger refinement time step of  $0.1 \pi$  s."

**Comment:** *P12, L9: Edit this sentence for better clarity. I suggest: "The results of benchmark tests 1 and 2 following the* 15 *Bochev et al. (2004) are shown in . . . "*

**Response:** We wanted to use the same naming convention as in the original paper (Example 1 and 2), so we now changed the sentence to read: "Our results of reproducing Examples 1 and 2 in the numerical results of (Bochev et al., 2004)...".

Comment: P12, L13: delete ", exemplary "

20 Response: Done.

**Comment:** P12, L16: You refer to the case of both refinement time steps. However, so far you have only talked about the single time step  $(0.1 \pi)$  that is used in Figure 6. I know that you are also talking about the  $0.01 \pi$  time step (shown in the sup docs), but it is not clear the way it is written right now. Please edit for clarity.

25 **Response:** We changed the sentence to: "The shapes of the concentration features are well recovered in the case of both refinement time steps (see Fig. 6b for refinement time step  $0.1 \pi$  s and Fig. A4b in the supplementary material for refinement time step  $0.01 \pi$  s)."

**Comment:** *P15*, *L22*: change "...glacier is becoming narrower..." to "...glacier becomes narrower..."

30 **Response:** Done.

**Comment:** *P16*, *L1-3*: *This sentence is too long and confusing and needs to be edited. I suggest deleting the unnecessary extra qualifiers:* ". . . *that uniformly cover wider portions of the accumulation area.* . . " *and "resulting in thick debris deposit but limited in area.* . . "

35 Response: We edited the sentence, it now reads: "In these simulations, ash fall or avalanche events that uniformly cover wider

portions of the accumulation area are included as layer-shaped debris deposits at the glacier surface. Rockfall events that result in a locally thick debris deposit are represented by a circular inclusion, as a possible remnant thereof. Both distinctly different debris inputs become severely elongated and band-like shaped during transport."

5 **Comment:** *P16, L2: Change "...inclusion as a possible..." to "...inclusion representative of a possible..."* **Response:** This sentence has been changed, compare previous comment.

**Comment:** *P18, L15: change start of line to* . . . *"that is as course as possible.* . . **Response:** Done.

10

30

**Comment:** *P18, L27: Start a new paragraph at this sentence.* **Response:** Done.

**15 4 Figure Comments**

We thank Anonymous Referee #2 for helpful comments to increase the comprehensibility of the figures. If there was more than one comment concerning the same figure, we labelled the referee's comments to be able to address them point by point.

**Comment:** Figure 1: Include a north arrow and scale bar

20 **Response:** Done.

**Comment:** Figure 3: - Please mark the ELA / beginning of the ablation zone. Although the reader can infer this from where the vertical velocity component passes zero from negative to positive, the addition of a simple arrow or line would be appreciated and aid in interpretation of Figure 9.

25 **Response:** As these glacier cases are based on fixed velocity fields (based on Eq. 1 in the manuscript and given glacier geometries) and are not connected to a mass balance routine, inclusion of an ELA marker could be misleading. We added a contour line of zero vertical velocity.

After contemplating the term "steady-state" as it has been used in the manuscript and the comment of Anonymous Referee #2 above, we came to the conclusion that the term is misleading in the absence of a mass balance model in our current study. Thus the term has been replaced with "fixed velocity field" in the manuscript.

**Comment:** Figures 3 and 4: - Since this geometry represents a glacier, but I suggest that you label x-axis and y-axis accordingly. i.e, "distance (m)" and "elevation (msl)

Response: We labelled the x-axis with "distance down-valley [m]" and the y-axis with elevation [m a.s.l.].

**Comment:** *Figure 5:*

1. I'm not sure why you have rotated the view individual panels unless you are trying to show the overall geometry. If this is the

5 case, then it is not effective but rather just makes the figure look rather messy and less clear. It is not required, but if possible, I suggest that you show all output in the same orientation.

2. Rather than using the axis labels x-axis, y-axis z-axis, consider using the physical interpretations for the labels (i.e. elevation, distance down-valley, distance cross-valley)

3. A wireframe showing just the glacial bed would be appreciated. As it is now, the reader has no way of knowing what the

10 subglacial topography looks like or where the bedrock "hump " is located.

4. This is out of main scope of the debris transport focus of the paper, but I am curious as to why there is such a pronounced positive vertical velocity component at the upper right side of the glacier near the valley bend. There is compressive strain encouraging the emergence of ice here, but the magnitude surprises me.

**Response:** 1. We now show all output in the same orientation and for clarity added two further panels. Panel (d) shows a 2D down-valley projection of the central flowline and panel (e) shows the underlying bed geometry.

2. Done.

3. We added this as panel (e).

4. This high compressive strain region which creates rather large but not unrealistic vertical velocities is a result of solving Eq. 1 (manuscript) for our assumed glacier geometry.

20

**Comment:** Figure 6: - The figure is fine. The caption could use some style adjustments for readability.

1. In this and all figure captions where you have separate panels, you do not need to use the phrase: "In (a) etc. . ... is shown " Just start with the intended panel letter and say what it is as a separate sentence.

2. Remove the sentence: "The data ranges from -0.14 to 1.11." We can see that from the figures.

25 3. Move the sentence "Color scales show concentration values" to the beginning or end of the caption.

4. Shorten the final sentence to: (f) Results of the convergence test as a function of mesh refinement parameter cvol. **Response:** 1. Done.

2. Done.

3. Done.

30 4. Done.

**Comment:** *Figure 7 caption:*

- Same style change comments as for figure 6.

Response: Done.

35

**Comment:** *Figure 8:**

- The first two panels fail to convey the intended information. I think that there are two separate surfaces represented for each contour (except 90) when I zoom way in, but it is barely possible to resolve these as separate at the print level of figure zoom. Also, which surface is the FEM solution and which is the analytical? These are the same color and are not labeled or marked.

5 I recognize that they are basically the same – which may be the point of the figure, but as it is now, it is just unclear. I suggest removing the panels (a) and (b) and just leaving in panel (c) which does convey usable information.

**Response:** In this figure, we intended to provide a visual comparison between the FEM and the analytical solution as an additional means of illustrating the model performance. We now adjusted the figure so that the contour lines of the FEM solution and the analytical solution can better be distinguished from each other, by showing the analytical solution in solid grey and

10 only the computed FEM solution in color and adjusted the figure caption accordingly, to highlight our intent with panels (a) and (b).

**Comment: Figure 9:**

1. The color scale (red gradient) for concentration is not effective at conveying any information as all the lines basically look

- 15 like the same shade of red. It may be impossible now, but if possible, consider changing this to a multiple-color colors scheme so that the change in concentration can actually be seen.
  - 2. Panels (a-c): Label the x-axis ('distance down-valley (m)). You probably only need to do this once for all panels.

3. Label (draw an arrow or something) the various debris layers (D1, D2, D3, etc.). Right now, it is not possible to track which is which layer.

- 20 4. Why not label the actual panel somewhere with their simulation times (24 yrs, 62 yrs, 85 yrs)
  - 5. Panel (d): label the x and y-axis correctly and put in at least two number values on each of the axes. Otherwise, the reader has no idea of the spatial scale.
  - 6. Why not label the actual panels with their simulation times (0.2 yrs, 8, yrs, 20 yrs, 26 yrs)

7. Caption: Remove unnecessary sentence: "Concentrations are displayed in the range of 0 to 100." You already show this on the color scale.

**Response:** 1. The concentration changes at the edges of the features are based on numerical diffusion only, which is well controlled, as demonstrated by our benchmark experiments and can be nicely seen in the cross-sections of Fig. 8c. Thus changing to a multiple-colors color scheme only creates very colorful edges but does not convey the results better and we consider it as visually less appealing (Fig. 2). As concentration is given as a continuous function and we do not want to introduce subjective

30 categories, we want to keep a sequential color scheme instead of a qualitative color scheme.

2. Done.

- 3. We labelled all the present debris inclusions.
- 4. Done.
- 5. Done.
- 35 6. Done.

7. As we do not want to misleadingly claim that our results have no numerical oscillations we wish to retain this sentence, but now rephrase it as: "Numerical oscillations as excursions beyond the initial values of 0 or 100 are of magnitude less than  $\pm 17$ and are truncated to the data limits".

**Figure 2.** Results of the debris transport simulations for the 2D longprofile of Haute Glacier d'Arolla for a multiple-colors color scheme (cf. Fig.9 in the manuscript). Debris concentration at 62 years after start of the simulations is shown. Concentrations are displayed in the range of 0 to 100, numerical oscillations as excursions beyond the initial values of 0 or 100 are of magnitude less than  $\pm 17$  and are truncated to the data limits.

**5 Comment: Figure 10:**

1. A rough 2-d outline of the glacier (surface, bedrock along the centerline in the x-z plane and the glacier sides in the x-y plane) would be helpful for interpretation. This does not have to be exact. Please label the axis relative to the glacier model (i.e. elevation, distance down-valley, distance cross-valley)

2. Label the debris distributions with their simulation times directly on the figure. There is plenty of room and this would make

**10 interpretation easier.**

3. Caption: Remove unnecessary sentence: "Concentrations are displayed in the range of 0 to 100." You already show this on the color scale.

**Response:** 1. We labelled the axes as suggested and added the glacier body in a transparent style in order to help to visually locate the debris inclusions.

15 2. Done.

3. As we do not want to misleadingly claim that our results have no numerical oscillations we wish to retain this sentence, but now rephrase it as: "Numerical oscillations as excursions beyond the initial values of 0 or 100 are of magnitude less than  $\pm 4$ and are truncated to the data limits".

20

**5 Supplementary Material Comments**

**Comment:** This may be a problem with my video player (I tried several) or my download, but I cannot play some of the .avi movie files. Please check that these are not damaged. These play successfully:

5 3d\_benchmark1.avi

2d\_glaciercase.avi These do not load (error): 2d\_benchmark\_exp1t20.avi 2d\_benchmark\_exp1t200.avi

10 2d\_benchmark\_exp2t15.avi 2d\_benchmark\_exp2t150.avi

> **Response:** We could successfully play all the downloaded videos with the following players on different operating systems: Ubuntu: VLC media player

Windows 7: VLC media player, Quicktime, Windows media player

15 macOS Sierra: VLC media player, Quicktime

**Comment:** *Figure A1:**

1. Please put a scale or tick marks on the x-axis and y-axis. the reader ha no idea what the scale is. Or is this dimensionless?

- 2. Label the color bar in panels (a) and (b).
- 20 *3.* In panel (b), if all the velocities are the same, then state that in the caption. Putting the "1.22" beneath the panel looks strange and does not convey any useful information.

**Response:** 1. We added in the text as well as the figure caption that the computations are performed and the data in the plots is shown for the unit square  $(1 \text{ m} \times 1 \text{ m})$ .

2. Done.

25 3. Done.

**Comment:** *Figure A2 and A3:**

1. Label the panels in the left hand column (2D concentrations) with the dt used in that row. The left hand panels need x-axis and y-axis labels/tick marks (0 - 1?), otherwise the reader has no way of knowing where the profiles in the middle and right

30 hand panels are taken from. You may also want to put two dashed lines across the concentration panels that show the location of the profiles.

2. It is odd that you put your References section in the middle of the document before Figure A2. Maybe this is something that happened in the auto-collate process during submission? I suggest just moving all Sup Doc references to the end of the document.

**Response:** 1. We labelled the axes and included lines to indicate the location of the right hand side profiles in the concentration plots. In order to prevent the figures from being crowded, we skipped the dt as a label.

2. Done.

**References**

Ackert, Jr., R. P.: A rock glacier/debris-covered glacier system at Galena Creek, Absaroka Mountains, Wyoming, Geogr. Ann. A, 80, 267–276, doi:10.1111/j.0435-3676.1998.00042.x, 1998.

Aschwanden, A., Bueler, E., Khroulev, C., and Blatter, H.: An enthalpy formulation for glaciers and ice sheets, J. Glaciol., 58, 441–457, 2012.

Bochev, P. B., Gunzburger, M. D., and Shadid, J. N.: Stability of the SUPG finite element method for transient advection–diffusion problems, Comput. Method. Appl. M., 193, 2301–2323, doi:10.1016/j.cma.2004.01.026, 2004.

Clark, D. H., Steig, E. J., Potter, Jr., N., and Gillespie, A. R.: Genetic variability of rock glaciers, Geogr. Ann. A, 80, 175–182, doi:10.1111/j.0435-3676.1998.00035.x, 1998.

10 Cuffey, K. M. and Paterson, W. S. B.: The Physics of Glaciers, Academic Press, Burlington, MA, 4th edn., 2010.

de Frutos, J., García-Archilla, B., John, V., and Novo, J.: An adaptive SUPG method for evolutionary convection-diffusion equations, Comput. Method. Appl. M., 273, 219–237, doi:10.1016/j.cma.2014.01.022, 2014.

- Fitzsimons, S. J., McManus, K. J., Sirota, P., and Lorrain, R. D.: Direct shear tests of materials from a cold glacier: implications for landform development, Quatern. Intern., 86, 129–137, 2001.
- 15 John, V.: A numerical study of a posteriori error estimators for convection-diffusion equations, Comput. Method. Appl. M., 190, 757–781, doi:10.1016/S0045-7825(99)00440-5, 2000.

Kirkbride, M. P.: Debris-Covered Glaciers, in: Encyclopedia of Snow, Ice and Glaciers, edited by Singh, V. P., Singh, P., and Haritashya, U. K., pp. 180–182, Springer Netherlands, Dordrecht, 2011.

Kowalewski, D. E., Marchant, D. R., Swanger, K. M., and Head, J. W.: Modeling vapor diffusion within cold and dry supraglacial tills of

- 20 Antarctica: Implications for the preservation of ancient ice, Geomorphology, 126, 159–173, doi:10.1016/j.geomorph.2010.11.001, 2011. LeVeque, R.: High-resolution conservative algorithms for advection in incompressible flow, SIAM J. Numer. A., 33, 627–665, doi:10.1137/0733033, 1996.
  - Mackay, S. L. and Marchant, D. R.: Dating buried glacier ice using cosmogenic 3He in surface clasts: Theory and application to Mullins Glacier, Antarctica, Quaternary Sci. Rev., 140, 75–100, doi:10.1016/j.quascirev.2016.03.013, 2016.
- 25 Mackay, S. L. and Marchant, D. R.: Obliquity-paced climate change recorded in Antarctic debris-covered glaciers, Nat. Commun., 8, 14 194, doi:10.1038/ncomms14194, 2017.
  - Mackay, S. L., Marchant, D. R., Lamp, J. L., and Head, J. W.: Cold-based debris-covered glaciers: Evaluating their potential as climate archives through studies of ground-penetrating radar and surface morphology, J. Geophys. Res.-Earth, 119, doi:10.1002/2014JF003178, 2014.
- 30 Monnier, S. and Kinnard, C.: Reconsidering the glacier to rock glacier transformation problem: New insights from the central Andes of Chile, Geomorphology, 238, 47–55, doi:10.1016/j.geomorph.2015.02.025, 2015.

Moore, P. L.: Deformation of debris-ice mixtures, Rev. Geophys., 52, 435-467, doi:10.1002/2014RG000453, 2014.

- Naito, N., Nakawo, M., Kadota, T., and Raymond, C. F.: Numerical simulation of recent shrinkage of Khumbu glacier, Nepal Himalayas, Proceedings of an International Workshop Held at the University of Washington in Seattle, Washington, USA, 13–15 September 2000,
- 35 IAHS Publication, 264, 245–254, 2000.
  - Reznichenko, N. V., Davies, T. R., and Alexander, D. J.: Effects of rock avalanches on glacier behaviour and moraine formation, Geomorphology, 132, 327–338, doi:10.1016/j.geomorph.2011.05.019, 2011.

Shroder, J. F., Bishop, M. P., Copland, L., and Sloan, V. F.: Debris-covered glaciers and rock glaciers in the Nanga Parbat Himalaya, Pakistan, Geogr. Ann. A, 82, 17–31, 2000.

---

## Author Response (AR2)

Anna Wirbel[1], Alexander Helmut Jarosch[2], and Lindsey Nicholson[1]

[1]Institute of Atmospheric and Cryospheric Sciences, University of Innsbruck, Innsbruck, Austria
[2]Institute of Earth Sciences, University of Iceland, Reykjavík, Iceland

We thank the Editor for his comments on our revised manuscript and split our response into three parts: (1) motivation for submission to TCD and TC, (2) comments on focus and relevance of the modelling tests on glaciers and (3) further rather minor points.

**1 Motivation for submission to TCD and TC**

5   We have been asked to comment on the concerns of the reviewers regarding the appropriateness of our submission to TCD and TC. The relevant reviewers comments are:

**Comment Reviewer 1 :** *With its focus on numerics and bench marking, this would be an exemplary submission to the EGU journal Geoscientific Model Development. As a contribution to TC Discussions it stands out as being mainly concerned with*
10 *a numerical model and, at this stage of progress, not much concerned with the Cryosphere. ... According to the contributors' guidelines for TC the journal invites "papers on all aspects of frozen water and ground on Earth" and "numerical modelling" is one of the main subject areas, I conclude that TCD is an appropriate target for this submission but it would be interesting to have wider discussion of this point because it could influence the future direction of the journal..*

15 **Comment Reviewer 2:** *My only general concern is related to the fit of this paper to the readers of The Cryosphere. Apart from the introduction, and the (understandably) preliminary 2-D and 3-D steady state glacier tests this paper contains very little that is relevant to the more general glaciology audience. The vast majority of the manuscript is concerned with the numerical model development, optimization, and benchmark testing. I am not criticizing the paper and think that it should be essentially published as is, but I will leave it to the editor to decide if it may be more appropriate for a different journal.*

In our initial response we did not explicitly respond to these comments as we felt they were either addressed directly to the Editor or indirectly to the editorial board whom oversee the direction of the Journal. Furthermore, the initial pre-assessment of the paper associated with acceptance for publication in TCD, stated that this paper was clearly within the scope of TC, and by sending the paper out for review, we assumed that the Editor had already accepted that the content was suitable for TCD and

consideration for TC. With apologies for not addressing these comments directly in our initial response to the reviews, we can summarize our position on choosing TCD as follows:

We accept that the article is a model development paper, and we did consider submitting to Geoscientific Model Development. However, as (1) the Aims and Scope of TCD, explicitly states that main subject areas include 'ice sheets and glaciers' and 'numerical modelling' of them, our contribution seemed to us to be entirely appropriate in the context of these guidelines, and (2) we preferred TCD as our model is intended to advance understanding of debris-covered glaciers, and we would therefore like to target readership and stimulate interest in our model from the cryospheric community.

**2    Comments on focus and relevance of the modelling tests on glaciers**

We thank the Editor for emphasizing the strengths of the presented model regarding tracking concentrations and the comments on improving the discussion of the glaciological applications. A new figure and detailed discussion thereof have been added in the Discussion Section to improve the manuscript in this respect. Concerning the model capabilities of tracking concentrations, we changed the paragraph in the Discussion starting on P17, L25 to: "The model presented here allows us to simulate the advection of debris concentration through a glacier in great detail. By explicitly modelling changes of debris concentration distribution as a result of transport within the glacier, local concentration changes (Eulerian perspective) caused by the deformation of debris deposit shape (Lagrangian perspective, cf. Fig.10) during englacial transport can be captured. Hence, for a given debris deposition event and glacier geometry, we can quantify the exact amount of debris concentration at any point in space and time. As a result, we can track the englacial debris transport and quantify the timing, location and debris concentration of a debris band emerging in the ablation zone, as well as quantifying how the location of maximum debris emergence from a debris band, and its dip angle, will change over time. This is all critical information for determining how the spatial pattern of surface debris thickness will develop and evolve in time.".

To comment on the points of clearer presentation of the performance regarding issues of numerical diffusion in the glacier cases and a relevant discussion thereof, we respond to the specific comments below.

**Comment:** *1) Adding (to Fig 9) some line-plot of profiles of concentrations for example across 2-3 debris features (bands, crevasse fills) before and after (and potentially during). You could plot these profile lines all in same figure with the centre of the feature in the middle of the cross-direction axis. This would further illustrate/quantify better how for example the shape (thickness) of such features has changed during transport.*

**Response:** We included a new figure in the Discussion (see Fig. 11 in the revised manuscript). This figure shows cross-profiles of debris features D1, D2 and the circular inclusion (labelled C0) from the 2D glacier test (see Sect. 4.2.1 and 5.2), with results shown in Fig. 9. We have chosen to focus on these features as they have long travel times deep inside the glacier and are thus subject to strong deformation as well as potentially to existing numerical diffusion. The cross-profiles are shown for time steps of $0.2$ years after the deposition of the respective feature (blue lines) and at the end of the simulation at $83$ years (red lines). The cross-profiles of the three features at the two different time steps are plotted on top of each other, for ease of interpretation.

A plot showing the exact location of the cross-profiles in Fig. 11 has been added to the supplementary material (Fig. A8). In the Discussion on P20 L01, the following paragraph describing the newly added Fig. 11 and the implications thereof has been added: " To demonstrate the concentration changes during transport within debris features for the 2D glacier test and its relation to numerical diffusion controlled by the cell area threshold (i.e. mesh refinement parameter), we plot concentration cross-sections for different debris features and model times in Fig. 11. The preservation of a central concentration peak over time is a good measure for identifying minimal numerical diffusion cases, as the redistribution of concentration by numerical diffusion would instantly lower that peak. In contrast, redistribution of debris concentration by transport preserves that peak due to the incompressibility of ice (cf. Sect. 1). Debris feature D1 experiences a strong amount of numerical diffusion over its 82.8 years of transport through the glacier and its central concentration decreases by $\sim 62$ %. This performance deficit is due to the choice of a cell area threshold being too large to preserve this initially thin debris feature. In contrast to D1, the central concentrations of debris features D2 and C0 are clearly well preserved (maximum concentration values above 100) and their change in concentration distribution (thinning of the cross-sectional length) is mostly due to transport. As a measure for debris band thickness over time we take full width at half maximum (fwhm) of the plotted debris concentration cross-sections in Fig. 11. D2 decreases from an initial fwhm of 9.3 m to a fwhm of 4.4 m over 57.8 years of transport. C0 decreases from an initial fwhm of 48.8 m to a fwhm of 2.2 m over 82.8 years of transport. Both cases clearly demonstrate the debris concentration changes caused by the transport through the glacier with minimal numerical diffusion. This highlights that a suitable cell area threshold choice is paramount to correctly simulate debris transport, as demonstrated with features D2 and C0. To improve the simulation results for feature D1, the only requirement is to lower the cell area threshold to a suitable value."

**Comment:** *2) Similar for Fig. 10, currently I really struggle to see in a 3D plot how well concentrations are preserved (remember tracking concentrations is the strength of the modelling method). So maybe add a plot similar to above, that shows a 1-d profile (along or across flow) of concentration through the debris feature (again with a relative spatial-coordinate, with the centre of the feature in the middle of the along or across-flow coord.). This would allow much better to demonstrate and see the behaviour and performance regarding variable debris concentration.*

**Response:** We want to emphasize that the idealized 3D glacier case is for demonstrating the 3D capabilities in a glaciological setting, and not to evaluate the model performance, which has been done in a formal manner for 3D in the 3D benchmark test (cf. Fig 8). The 3D benchmark test (see Sect. 4.1 and 5.1) shows that a similar concentration feature (with decreasing concentration values towards its edges), subjected to an even more complex velocity field, can be well simulated with the given model and an appropriate mesh setup. For the idealized 3D glacier case, it is not possible to quantify numerical diffusion from the test case we present here. The only possibility to assess numerical diffusion in such a setting is to run additional tests with different mesh refinement parameters, as (a) an analytical solution for such a case and (b) appropriate field measurements are missing. A figure showing cross-profiles of concentration similar to the newly added Fig. 11 for the 2D glacier test, is difficult to interpret as in the idealized 3D glacier test, the initial concentration has been described as a smoothed function at the boundaries of the feature (cf. Fig. 10 debris feature labelled "0 yrs"). Thus, compared to the 2D glacier test in Fig. 9, concentration should not be 100 throughout the feature and during the entire transport, as the concentration initially has not been set to 100 everywhere

within the feature. This implies that the observed decrease in concentration towards the edges of the feature during transport is not only due to numerical smearing, but also a result of the initially chosen debris concentration distribution. Therefore, in this case the preservation of the initial concentration peak ($c = 100$) is the only measure to quantify numerical diffusion. We attempt to visualize this property in Fig. 1, where the preservation of the central concentration for $c \geq 90$ is visualized as red

5   isosurfaces at $c = 90$. The first 30 years of transport, this concentration feature ($c \geq 90$) is preserved, however afterwards the whole concentration feature starts to leave the glacier and we do not consider later timesteps in this analysis. Even though Fig. 1 demonstrates that numerical diffusion is sufficiently low to maintain a concentration $c \geq 90$ at the centre of the feature, we do not want to include this analysis in the main manuscript as we do not want to give recommendations of mesh refinement parameters for glaciological settings in general. Hence we state in the manuscript on P21 L3 that for each application a suitable

10  mesh refinement parameter needs to be determined. Furthermore Fig. 10 already displays cross-cuts through the debris features at different times during transport visualized as isosurfaces of concentration and colour code. The deformation of the feature shape can be seen, but by cutting through the features also the internal changes/redistribution of concentration are visualized. Due to the demonstration nature of this case, we did not emphasise the visualization of concentration distribution changes to avoid confusing this demonstration with an evaluation of the model capabilities (which have been extensively demonstrated in

15  our benchmark tests, where the behaviour and performance regarding variable debris concentration is visualized with isosurfaces of the concentration at different time steps as well as cross-sections of the analytical solution and the model simulations at the final time step).

[Figure]

**Figure 1.** Debis concentration over time, with $c = 1$ as opaque blue isosurfaces and $c = 90$ in red. Concentrations are plotted for the times $t = 0, 10, 20, 30$ years along the principal flow direction from left to right.

**3 Further rather minor points**

**Comment:** *P. 6 line 13; I assume the numbers in the brackets refer to min and max, why not say this more explicitly e.g. (min 18.2, max. 1.4).*

**Response:** We included the min. and max. statements in the brackets.

**Comment:** *p. 11, section 4.2.2.: I think this 3D surface-geometry requires some more explanation where it is from, or how it has been constructed. I ask this because you want an as realistic as possible geometry and related flowfield. Is it a result of some build up (spinning up) experiment, or even related to a real world case?*

**Response:** We fully appreciate the comment of the Editor, we do agree with the Editor that a highly realistic flow field is required for any type of benchmark tests, however we do not agree that our current presentation of the model capabilities in 3D requires "as realistic as possible geometry and related flow-field". Again, we want to emphasize that the idealized 3D glacier case is for demonstrating the 3D capabilities in a glaciological setting only, and not to evaluate the model performance. In principle, to demonstrate the 3D capabilities of the model for a glaciological setting, only a simple geometry representing "a glacier" at the same scales of a real glacier is required. Therefore, we generated a simple glacier geometry that is not based on a real world case but has plausible dimensions for a glaciological application. Having demonstrated extensively the capabilities of our transport model to solve Eq. 5a in our numerical benchmark tests (Sect. 4.1 and 5.1) we have very high confidence that the transport model can deal with any complex flow fields as input.

**Comment:** *p. 17: here the layer shaped features are mentioned, BUT NOT SHOWN, which apparently deform to arcuate concave features. It is a bit tricky for the reader to digest, and just not clear why they are not shown. I would either show it (maybe in appendix) or leave this sentence away as the reader has no way of judging this without evidence.*

**Response:** We skip this sentence in the revised manuscript, as we chose to not present initial layer results at this stage as they deserve a detailed discussion and would be more informative if they were based on a real world example. We plan to perform such a study in the near future.

**Comment:** *p. 17, line 15: i am a bit critical about rockfall events producing thick debris deposits (sheprical). Maybe this applies for very small rockfalls, but the ones I am aware off or have seen in field are rather relatively thin extended sheets (layers). Maybe rephrase this a bit.*

**Response:** We wanted to show an end-member case of debris deposits compared to the thin layer-shaped features, this is why we chose a circular inclusion. We rephrased the sentence on P 17, L 15 to: "Rockfall events that result in a locally thick debris deposit are represented by a circular inclusion, as an end-member case of possible remnants thereof." Similar to the geometry chosen for the 3D glacier case, here the choice of debris inclusion shape is motivated by demonstrating the model capabilities and not to represent the most realistic conditions encountered in the field.

**Comment:** *P. 17 line 23-25: this is more a note than an urgent requirement to change anything. Yes I agree that this results in a non-uniform debris thickness, but if one considers that the debris once melted out at the surface and just advected on the surface downstream one probably gets a surprisingly smooth thickness distribution at least in an along flow sense and after some time. So 'uniform' debris thickness may not be so wrong. Maybe this could be formulated a bit more nuanced.*

**Response:** As both field and satellite determinations of debris thickness indicate that debris thickness is typically non-uniform over the glacier surface (e.g. Mihalcea et al., 2008; Zhang et al., 2011), and as the location of discrete zones of debris melt-out or deposition are critical to existing conceptual models of debris cover development (e.g. Anderson, 2000; Kirkbride and Deline, 2013), we consider this point an important one to make. Especially as the ability to quantify this aspect of debris cover development is one of the key advances our model offers.

A certain situation of debris supply, melt-out, flow velocity and advection could arise such that over time the debris cover thickness is relatively uniform, but at present it is not known how likely or common such a situation would be. We have adjusted the sentence on P17 L22 to read "Hence, an assumption of a uniform englacial debris distribution of constant englacial debris concentration (Naito et al., 2000) that would result in a continuously debris-covered glacier surface where surface ablation is occurring, might not reflect reality adequately in order to capture the geometrical response of the glacier to the developing debris cover." We capture the tentative nature of this statement by using "might not reflect reality", and indeed we hope that future simulations with our coupled glacier-debris model will help identify the degree to which the spatial pattern of emergence of debris plays a role in the glacier evolution.

**Comment:** *p.19 last line/p20 first line: '..but are still suitable for glaciological application'. Yes this may be the case but this applicability (accuracy, perforambce in concentration tracking) should be better shown/visualized/demonstrated (see main comment above)).*

**Response:** We now explicitly state that the suitability of the model for glaciological applications is a conclusion we draw based on its performance in benchmark tests with complex flow fields. The sentence on P20 L18 now reads: "The results we show here in the case of an idealized 3D glacier geometry will be subject to some numerical smearing, however the 3D benchmark test shows that numerical smearing can be minimized by choosing a suitable mesh refinement parameter."

**Comment:** *Fig. 1: the 'grey' line is very hard to see, if it is important make it clearer (maybe black and thicker).*

**Response:** Done.

**Comment:** *Fig. 4 caption line 2: '...the location and extent of the additional debris layers deposited later at the surface at the indicated times.'*

**Response:** We changed line 2 of the caption to: "The horizontal lines indicate the location and extent of the additional debris layers deposited at the surface at the stated times. L indicates the horizontal distance of the glacier surface where debris is deposited."

**Comment:** *Fig. 5: (d) I personally find the 3D-profile plot along the flow line (d) not very helpful as it is '3d'- again, its is very hard to quantify anything. Why not just plot a 2-dim profile along the flowline in along-flow and z dimensions and then indicate the flowline in 5(a) (e) at least in my printed copy it is very hard to see any bed structure in this figure, as it appears just as a blue blob. Can this be improved, maybe plot thicker 'wires' or add colours as elevation?*

**Response:** We adjusted the figure and now show a 2D profile of the flowline (Fig. 5d) that is indicated in Fig. 5e. We adjusted the colours in Fig. 5e for better visibility of the bedrock wires and also show the bedrock elevation in colour.

**Comment:** *Fig. 9: add perhaps some additional figure here that shows zoomed in profile plots visualizing concentration (changes, diffusion) across some features (see main comments)*

**Response:** See response in point 2.

**Comment:** *Fig. 10: again add perhaps some additional figure here that shows zoomed in profile plots visualizing concentration (shape change, concentration) across some features (see main comments)*

**Response:** See response in point 2.

[revised manuscript text omitted]

---

## Author Response (AR3)

Anna Wirbel[1], Alexander Helmut Jarosch[2], and Lindsey Nicholson[1]

[1]Institute of Atmospheric and Cryospheric Sciences, University of Innsbruck, Innsbruck, Austria
[2]Institute of Earth Sciences, University of Iceland, Reykjavík, Iceland

We thank the Editor for this final comment and would like to take the opportunity and again thank both Referees and the Editor for their time and valuable input that have improved the manuscript.

**1   Minor point**

**Comment:** *Fig11 caption: I somewhat struggled initially how these profiles are oriented (and had to consult fig A8) and how they are meant. I would help the reader here by say that they are 'vertical': 'Vertical cross-profiles at different models times through the debris features D1,.... The vertical profiles ....'. You may also clarify that the absolute position on the x-axis is somewhat arbitrary but that for better comparison, the same features have for different times be plotted with the centre at the same x-position*

**Response:** Thanks for this comment, we adjusted the caption of Fig. 11 to read: " 
[revised manuscript text omitted]